



# The potential role of organics in new particle formation and initial
# growth in the remote tropical upper troposphere
Agnieszka Kupc[1,2], Christina J. Williamson[1,3], Anna L. Hodshire[4], Jan Kazil[1,3], Eric Ray[1,3], T. Paul, Bui[5],
Maximilian Dollner[2], Karl D. Froyd[1,3], Kathryn McKain[3,6], Andrew Rollins[1], Gregory P. Schill[1,3],
Alexander Thames[7], Bernadett B. Weinzierl[2], Jeffrey R. Pierce[4] and Charles A. Brock[1]
[1]Chemical Sciences Laboratory, National Oceanic and Atmospheric Administration, Boulder, CO 80305, U.S.A
[2]Faculty of Physics, Aerosol Physics and Environmental Physics, University of Vienna, 1090 Vienna, Austria
[3]Cooperative Institute for Research in Environmental Sciences, University of Colorado, Boulder, CO 80309, U.S.A.
[4]Department of Atmospheric Science, Colorado State University, Fort Collins, CO 80523, USA
[5]Earth Science Division, NASA Ames Research Center, Moffett Field, California, USA
[6]Global Monitoring Laboratory, National Oceanic and Atmospheric Administration, Boulder, CO, 80305, USA
[7]Department of Meteorology and Atmospheric Science, Pennsylvania State University, University Park, PA, USA
*Correspondence to*: Agnieszka Kupc ([agnieszka.kupc@univie.ac.at](mailto:agnieszka.kupc@univie.ac.at))
**Abstract.**
Global observations and model studies indicate that new particle formation (NPF) in the upper troposphere (UT) and
subsequent particles supply 40-60 % of cloud condensation nuclei (CCN) in the lower troposphere, thus affecting the Earth's
radiative budget. There are several plausible nucleation mechanisms and precursor species in this atmospheric region, which,
in the absence of observational constraints, lead to uncertainties in modeled aerosols. In particular, the type of nucleation
mechanism and concentrations of nucleation precursors, in part, determine the spatial distribution of new particles and resulting
spatial distribution of CCN from this source. Although substantial advances in understanding NPF have been made in recent
years, NPF processes in the UT in pristine marine regions are still poorly understood and are inadequately represented in global
models.
Here, we evaluate commonly used and state-of-the-art NPF schemes in a Lagrangian box model to assess which
schemes and precursor concentrations best reproduce detailed in situ observations. Using measurements of aerosol size
distributions ($0.003 < Dp < 4.8$ µm) in the remote marine troposphere between ~0.18 and 13 km altitude obtained during the
NASA Atmospheric Tomography (ATom) mission, we show that high concentrations of newly formed particles in the tropical
UT over both the Atlantic and Pacific oceans are associated with outflow regions of deep convective clouds. We focus analysis
on observations over the remote Pacific Ocean, which is a region less perturbed by continental emissions than the Atlantic.
Comparing aerosol size distribution measurements over the remote Pacific with box-model simulations for 32 cases shows
that none of the NPF schemes most commonly used in global models, including binary nucleation of sulfuric acid and water



(neutral and ion-assisted) and ternary involving sulfuric acid, water, and ammonia, are consistent with observations, regardless
of precursor concentrations. Through sensitivity studies, we find that the nucleation scheme among those tested that is able to
explain most consistently (22 of 32 cases) the observed size distributions is that of Riccobono et al. (2014), which involves
both organic species and sulfuric acid. The method of Dunne et al. (2016), involving charged sulfuric acid-water-ammonia
nucleation, when coupled with organic growth of the nucleated particles, was most consistent with the observations for 5 of
32 cases. Similarly, the neutral sulfuric acid-water-ammonia method of Napari (2002), when scaled with a tuning factor and
with organic growth added was most consistent for 6 of 32 cases. We find that to best reproduce both nucleation and growth
rates, the mixing ratios of gas-phase organic precursors generally need to be at least twice that of $SO_2$, a proxy for dimethyl
sulfide (DMS). Unfortunately, we have no information on the nature of oxidized organic species that participated in NPF in
this region. Global models rarely include organic-driven nucleation and growth pathways in UT conditions where globally
significant NPF takes place, which may result in poor estimates of NPF and CCN abundance and contribute to uncertainties
in aerosol-cloud-radiation effects. Furthermore, our results indicate the organic aerosol precursor vapors may be important in
the tropical UT above marine regions, a finding that should guide future observational efforts.

## 1 Introduction

The majority of particles found in the atmosphere are formed through gas-to-particle conversion (i.e. nucleation) from
clustering of low-volatility vapors (Gordon et al., 2017; Pierce, 2017). While the formation of these molecular clusters appears
to take place almost everywhere and at all times in the atmosphere (Kerminen et al., 2018), the formation of thermodynamically
stable aerosol particles with diameters ($D_p$) ≳1.5 nm requires favorable conditions in terms of temperature, availability of
condensable vapors, and the background of pre-existing bigger particles that compete for condensing vapors, and so may not
occur in every atmospheric environment (Kulmala et al., 2014). Most of these newly formed particles are lost by coagulation
with larger particles, and do not contribute to particle number (Westervelt et al., 2014). A subset of the nucleated particles
grows by condensation to become larger particles with reduced Brownian motion, and hence lower coagulational loss rates
(e.g., Pierce and Adams, 2007). Particles with $D_p$ ≳50 nm can serve as CCN at supersaturations found in typical marine
cumulus and stratocumulus clouds (Quinn et al., 2008), increasing droplet number concentrations and cloud albedo, and thus
indirectly affecting the Earth's radiative budget (Twomey, 1974; IPCC, 2013).
The tropical UT is known to be a major source region of new particles (e.g. Clarke, 1993; Brock et al., 1995; Clarke
and Kapustin, 2002; Weigel et al., 2011; Williamson et al., 2019). This strong aerosol production is believed to be linked with
frequent deep convection in this region. The mechanism proposed by Clarke (1992) involves the formation of new particles in
the UT from convectively lifted and cloud-processed boundary layer air. At the conditions of cold temperatures, high photolytic
fluxes, and low concentrations of pre-existing aerosol particles found in the outflow of deep convection at altitudes >8 km,
aerosol precursor gases that may survive convective transport and scavenging can oxidize and nucleate new particles which
then grow to CCN sizes as they descend in the gradually subsiding air that compensates for the upward convection. Raes


(1995) used a box model to determine that observed concentrations of CCN in the remote marine boundary layer (MBL), and
their temporal stability, could not be explained without a source of particles being entrained from the free troposphere (FT).
Clarke et al. (2006) estimated that entrainment from the FT provides 35-80 % of the CCN flux into the MBL over latitudes
between 40° S and 40° N with the rest coming from sea salt aerosol. More recently, Quinn et al. (2017) found that at ~0.5 %
supersaturation, the accumulation mode aerosol, composed primarily of sulfate compounds rather than sea-spray particles,
provides ~70 % of the CCN population throughout the MBL of the tropics and midlatitudes, and suggested that these particles
originate from the FT.
Despite the climatic importance of NPF in the tropical UT, the chemical mechanisms are poorly understood (e.g.,
English et al., 2011).  This lack of understanding is driven by the fundamental complexity and variability of the atmosphere,
the range of potential chemical species and mechanisms that could lead to NPF and subsequent growth of the newly formed
particles to CCN, and the difficulty in obtaining observations of processes occurring in remote areas, at high altitudes, and
over time scales ranging from minutes (NPF) to weeks (condensational growth during gradual descent). Together, these issues
have made it difficult to validate NPF schemes used in global models and have hindered our ability to reduce uncertainty in
aerosol-cloud-radiation interactions.
Williamson et al. (2019) showed that three of four global models examined in their study underestimated the
magnitude of NPF in the tropical Pacific UT and all failed to accurately simulate the abundance of CCN-sized particles in the
lower troposphere of the same region (the fourth model significantly overestimated aerosol loadings throughout the
troposphere). None of these models used a NPF scheme involving organics, and the three models may lack sufficient precursor
vapors for growth, in addition to other deficiencies. Previous model studies (e.g., Kazil et al., 2010; Yu et al., 2010; Zhang et
al., 2010; Zhu et al., 2019) show that the choice of NPF mechanism can drive substantial changes in the predicted abundance
and spatial distribution of particles. While Westervelt et al. (2014) suggested that the global-mean boundary layer CCN are
not very sensitive to the number of particles formed in the UT due to the dampening effects of coagulation (i.e., more nucleation
leads to faster coagulational losses), different choices of NPF mechanisms in models might alter the spatial and temporal
pattern of NPF, and thus affect the spatial distribution and magnitude of CCN abundance. It is clear that accurate simulation
of NPF and growth processes is essential to adequately represent particle size distributions and their spatial distribution in
global models and improve predictions of aerosol-cloud-radiation effects (Hodshire et al., 2018; Williamson et al., 2019).
Field measurements have shown that sulfuric acid is a key component in atmospheric NPF in the continental boundary
layer (e.g., Weber et al., 1997; Riipinen et al., 2007; Sihto et al., 2006). Several nucleation schemes involving sulfuric acid
have been used in global models as a consequence. These include activation nucleation scheme that depends on sulfuric acid
only (Kulmala et al., 2006), binary schemes that involve sulfuric acid and water to form new particles (e.g., Vehkamäki et al.,
2002), or ternary schemes in which sulfuric acid, water and ammonia condense to form new particles (e.g., Napari et al. 2002).
The activation nucleation scheme, however, is an empirical formulation tuned to mid-latitude continental boundary layer
observations so it is appropriate to use only there. Binary NPF has been suggested to be favored in the remote tropical UT due
to cold temperatures, high relative humidity (RH), and the availability of supersaturated sulfuric acid (Clarke, 1992; Brock et





al., 1995; Clarke and Kapustin, 2002). Ion-assisted nucleation of sulfuric acid and water clusters has been identified as a
potential pathway for binary NPF (Kirkby et al., 2011; Lovejoy et al., 2004; Kazil and Lovejoy, 2007; Raes et al., 1997; Yu
2010). Ions stabilize the molecular clusters so that nucleation can occur at warmer temperatures and lower nucleating-vapor
concentrations (Yu, 2010).

Recent observations of the composition of molecular clusters present during NPF have highlighted the role that
organics may play (Kulmala et al., 2013; Smith et al., 2004). Murphy et al. (2006) and Froyd et al. (2009) found that larger
particles ($D_p$> 0.15 µm) in the UT contained significant organic matter that was likely secondary, which suggests that
condensable gas-phase organic compounds are present in the UT. Andreae et al. (2018) postulated that biogenic volatile organic
compounds carried from the boundary layer to the UT by deep convection and oxidized to form condensable species over the
Amazon are responsible for NPF observed in this continental UT region. Weigel et al. (2011) also suggested that organics
might contribute to NPF events observed in the UT. Other nucleation processes combining sulfuric acid with ammonia (Kürten
et al., 2016; Merikanto et al., 2007), amines (Almeida et al., 2013), di-amines (Jen et al., 2016), or organics (Kulmala et al.,
2006; Metzger et al., 2010; Riccobono et al., 2014), or organics alone (Kirkby et al., 2016; Bianchi et al., 2016), have been
proposed to explain some field and laboratory observations of NPF, primarily at warmer temperatures and continental
locations. In a modeling study, Zhu et al. (2019) found that pure organic nucleation from biogenic volatile organic compounds
could be an important source of particles, especially in the UT of modern-day pristine, continental environments and during
the pre-industrial period.

Because there have been no in situ observations of the composition of molecular clusters and nano-particles found in
convective outflow in the UT, it is difficult to ascertain which of these varied mechanisms, if any, contribute to NPF in the
remote FT. In this study, we use unique observations obtained during the Atmospheric Tomography Mission (ATom), a multi-
year airborne program to measure gas and aerosol properties of the remote troposphere over both the Atlantic and Pacific
oceans across four seasons. Recently formed particles observed in the tropical UT were linked to recent outflow from deep
convection. We use box models constrained by trajectory calculations to evaluate how well different NPF formation
mechanisms can simulate the observed particle size distributions. We perform extensive model sensitivity studies to determine
which nucleation mechanisms and initial precursor mixing ratios allow for the model to match observed size distributions.

## 2. Methods

To establish a link between convection and NPF, and to explore the processes that govern NPF and initial growth in
the tropical and subtropical free troposphere over the Pacific Ocean, we couple measured size distributions between 2.6 nm
and 4.8 µm in diameter from the four ATom missions with calculated air mass back trajectories and two box models. The back
trajectories identify air masses potentially influenced by recent convection. We compare our simulations with in-situ ATom
observations of aerosol size distributions. We examine which nucleation schemes best explain the observations, and evaluate
whether observed sulfur precursors (SO$_2$ and dimethyl sulfide (DMS)) can explain the NPF and the particles' initial growth.





In one model, we simulate particle formation by neutral and charged binary and ternary schemes, and a neutral organic scheme,
and we also add organics for initial growth of the particles in all schemes. In an additional model, we form particles using both
neutral and charged binary schemes.

**2.1 The Atmospheric Tomography Mission**

The NASA Atmospheric Tomography Mission (ATom) was an airborne global survey that used the NASA DC-8

research aircraft to map for the first time the composition of the remote atmosphere over both the Pacific and Atlantic basins
(~82° N to ~86° S; Fig. 1) in nearly continuous ascents and descents between ~0.18 and 13 km in all four seasons (July-August
2016, January-February 2017, September-October 2017, and April-May 2018). The primary objectives of the mission were to
examine the composition of the remote atmosphere to improve understanding of photochemical mechanisms for reactive and
long-lived gas-phase species and to identify the abundance, distribution, sources of aerosol particles in the remote marine
troposphere (Wofsy et al., 2018).

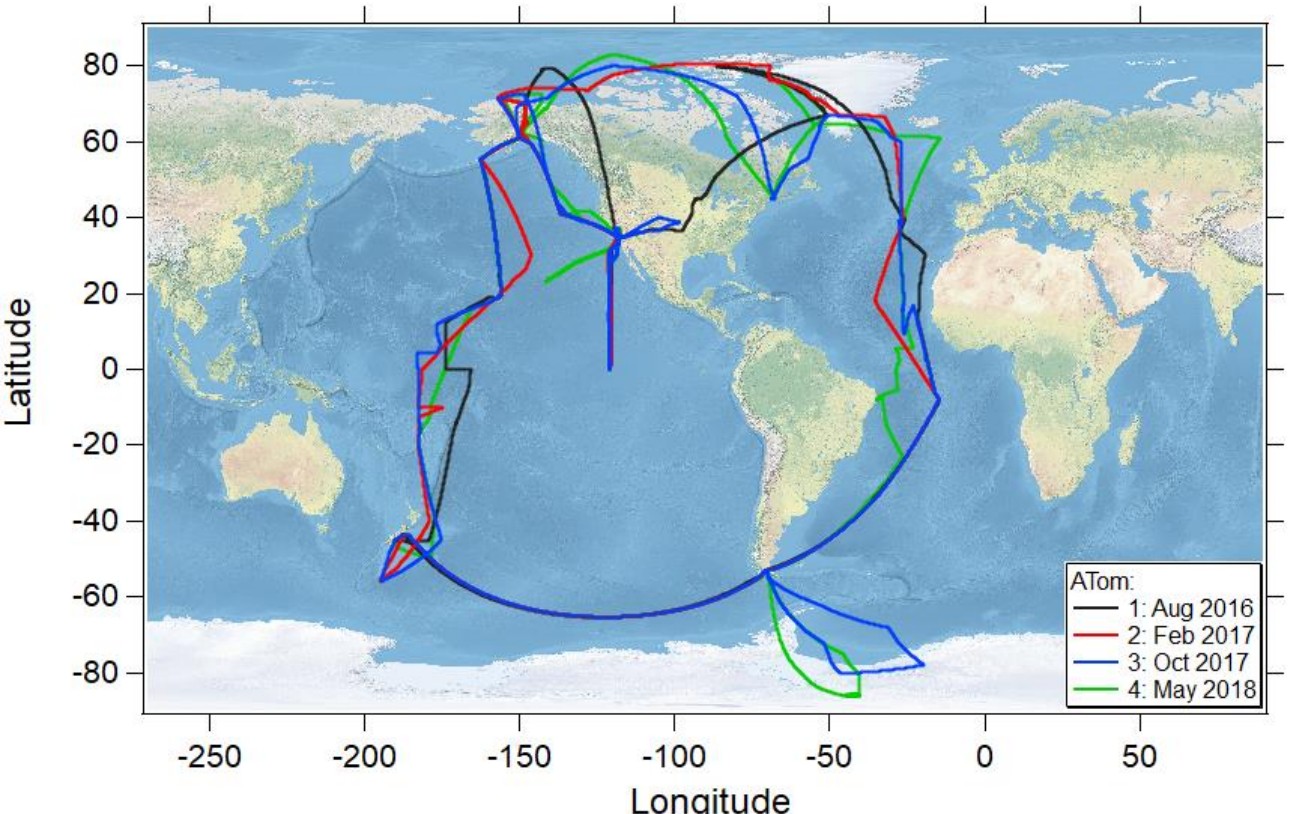


**Figure 1: Flight tracks of the NASA DC-8 research aircraft during the ATom 1 (July-August 2016), ATom 2 (January-February**
**2017), ATom 3 (September-October 2017) and ATom 4 (April-May 2018) missions covering the remote marine atmosphere of the**
**Pacific and Atlantic Oceans between ~83⁰ N and ~86⁰ S.**



## 2.1 Measurements

A suite of fast time response (1 Hz) particle counters and optical particle size spectrometers were used to measure dry size distributions between 2.6 nm and 4.8 µm in diameter (Brock et al., 2019). Two nucleation mode aerosol size spectrometers (NMASS; Williamson et al., 2018), each consisting of five continuous-flow condensation particle counters (CPCs) with different fixed cut-off diameters (i.e. diameters at which each CPC detects 50 % of the incoming particles) between 3.2 and 59 nm, measured particle number concentration. Two optical aerosol counters, an ultra-high-resolution sensitivity aerosol spectrometer (UHSAS; Kupc et al., 2018) and a laser aerosol spectrometer (LAS; Froyd et al., 2019), measured particle size distributions from $0.06<D_p<1$ µm and $0.12<D_p<4.8$ µm, respectively. All concentrations reported here are corrected to standard temperature and pressure (STP), 1013 hPa and 273.15 K. The NOAA Particle Analysis by Laser Mass Spectrometry (PALMS) instrument measured the composition of individual aerosol particles (Froyd et al., 2009; 2019). For this study the PALMS size range is restricted to 0.125-1.5 µm. Due to inlet sampling artifacts (Weber et al., 1998; Murphy et al., 2006), cloudy periods were removed from the analysis. Clouds were detected using a second-generation cloud, aerosol, and precipitation spectrometer (CAPS) mounted under the wing, which also measured coarse aerosols >0.5 µm (Dollner et al., in preparation) at ambient conditions.

Temperature, pressure, and wind speed with high time resolution (1 Hz) were measured with an accuracy of ±0.3 K, ±0.3 hPa, and ±1.0 m s$^{-1}$, respectively (Scott et al., 1990). Highly sensitive sulfur dioxide (SO$_2$) measurements were made during ATom 4 using laser-induced fluorescence (Rollins et al., 2016; Rollins et al., 2017) with a precision (1σ) of 1-2 pptv at 10 s and an overall uncertainty of ±(9 % +2 pptv). Laser-induced fluorescence was used to measure OH and HO$_2$ simultaneously (Faloona et al., 2004; Brune et al., 2020) with an accuracy of ±35 %. Measurements of carbon monoxide (CO) were made using a Picarro G2401m (Chen et al., 2013) with a precision (1σ) of 2-3 ppb at 10 s and an average uncertainty of 4 ppb. All data used in this analysis can be found in Kupc et al. (2020).

## 2.2 Air mass back trajectories and convective influence

To identify air in the UT influenced by recent deep convection, we calculated 10-day air mass back-trajectories using the Bowman trajectory model (Bowman, 1993) driven with meteorological fields (3 hourly, 0.25˚ horizontal resolution) from the National Centers for Environmental Prediction (NCEP) Global Forecast System (GFS). Trajectories were also run with MERRA2 and ERA5 reanalysis meteorology and the results were similar. Meteorological products from NCEP interpolated to the aircraft flight track agreed best with quantities measured on the aircraft during ATom, so all analyses were done using the trajectories based on the NCEP data.

A cluster of 245 trajectories was initialized within a grid (0.3˚ x 0.3˚ x 20 hPa; Fig. S1) centered around the DC-8 flight track location every minute of flight (Fig. 1). The back-trajectory time step was 3 hours, based on the reanalysis data, while a time step of 15 minutes interpolated from the 3 hours reanalysis data was used for box model simulations. Uncertainty in the back-trajectory locations is represented by the 3-D spread in the trajectory cluster. The vertical uncertainty is estimated



as the standard deviation in pressure (hPa) of the trajectory cluster at each time. The horizontal uncertainty is estimated using
a probability grid based on the trajectory cluster in longitude and latitude at each time (Fig. S1), where probability grid refers
to the number of trajectories at each time that are within each latitude-longitude grid box (2˚ x 2˚). For instance, if 24 of the
trajectories are within a certain grid box at a certain time then the probability for that grid box is ~10% (24/245). The probability
that air sampled by the aircraft was influenced by deep convection was calculated based on coincidences of the back-trajectory
cluster with satellite derived cloud locations and characteristics such as cloud top and base pressure levels (NASA Langley).
To isolate deep convection, only clouds with vertical extent >5 km were considered in the convective influence (CI)
calculation. The CI probability is the fraction of the trajectories in each cluster that intersected a convective system within a
distance tolerance of 0.15˚ (~10-15 km), and for which the RH with respect to liquid water ($RH_w$) of the trajectory was >50
%. If the CI probability determined in this manner was >95%, we assume that the aircraft was sampling air strongly influenced
by deep convection.
**2.3 Description of models**

We use two independent aerosol nucleation and growth box models to test if different nucleation schemes are

consistent with observations, following the trajectories from convective outflow to the location of the aircraft. These two
models are conceptually similar, but differ in size resolution and their support for different nucleation mechanisms. Our
primary model, the TwO-Moment Aerosol Sectional (TOMAS; Adams and Seinfeld, 2002; Pierce and Adams, 2009; Pierce
et al., 2011) includes both neutral and charged mechanisms. The neutral mechanisms include sulfuric acid and water (binary
scheme; Vehkamäki et al. (2002)), sulfuric acid, water, and ammonia (ternary scheme; Napari et al. (2002)), and sulfuric acid
with organics (organics scheme; Riccobono et al., 2014; Yu et al., 2017). The charged mechanism is Dunne et al. (2016),
which quantifies NPF in terms of sulfuric acid, ammonia, and ion concentration (also including neutral pathways). In addition
to testing the role of organics in nucleation and growth, we also test the influence of organics on aerosol initial growth when
added as a condensing species following the nucleation of particles formed by each of the nucleation schemes in TOMAS, as
described in Sect. 2.3.1.

We also use the Model of Aerosols and Ions in the Atmosphere (MAIA; Lovejoy et al., 2004; Kazil and Lovejoy,

2007) to test ion-assisted nucleation of sulfuric acid and water. Since ion-assisted nucleation simulations using MAIA did not
explain the observed size distributions in our work, we focus on TOMAS model description and results, and the details on the
MAIA model are in the Supplemental Material (Section S1). Some features common to both models are described below.

The MAIA and TOMAS box models are constrained to follow the meteorological conditions along the trajectories.

They are initialized at the point where the trajectories intersect deep convection, and proceed forward in time until reaching
the aircraft sampling location and time. The temperature, pressure, and $RH_w$ vary based on the trajectory. We vary the initial
$SO_2$, $NH_3$, and organic aerosol precursors in TOMAS and $SO_2$ in MAIA (Table 1 and Supplemental Material Table S1) to see
which initial values of these species allow for the best matches to the observed size distribution. We note that neither model
explicitly simulates DMS, which is likely to be an important aerosol precursor through its oxidation to form $SO_2$ and





subsequently H$_2$SO$_4$, as well as through its oxidation to methanesulfonic acid (MSA; Hodshire et al., 2019), which is a
condensing species that may also be able to participate in NPF (Bork et al., 2014; Chen et al., 2015; Chen and Finlayson-Pitts,
2017). Previous analyses have shown that most of the observed reactive gas phase sulfur above the boundary layer is in the
form of SO$_2$ (Veres et al., 2019). In this work, both models are initialized with a measurement-based, pre-existing background
aerosol population that acts as a sink for condensable vapors and small particles (see Section 2.4). Nucleation-mode particles
are initialized at zero concentration. We calculate the OH diurnal cycle using a prescribed peak noontime value based on
observations of OH on the DC-8 aircraft (Section 2.4 and Supplemental Material Fig. S3). The OH concentration along the
trajectory and the resulting production rate of H$_2$SO$_4$ from oxidation of SO$_2$ are then calculated. We ignore possible enhanced
OH due to cloud reflectivity in the vicinity of convective outflow and reduced OH from shading by higher clouds.

**2.3.1 The TOMAS box model**

The TOMAS model simulates particle nucleation, condensation, and coagulation in 43 logarithmically spaced particle
size bins, which represent dry diameters from 0.7 nm-10 µm. TOMAS tracks the total aerosol number and mass of each species
for each size bin. The simulated aerosol species are sulfate, ammonia, a representative oxidation product of biogenic organics,
and water. In these simulations, neutral sulfuric acid-water nucleation is based on Vehkamäki et al. (2002), neutral sulfuric
acid-water-ammonia nucleation is from Napari et al. (2002), ion-induced and neutral sulfuric acid-water and sulfuric acid-
water-ammonia nucleation is from Dunne et al. (2016), and neutral sulfuric acid-organic nucleation is from Riccobono et al.

(2014).

Vehkamäki et al. (2002), referred to here as VEHK, describe a parametrization for neutral sulfuric acid-water particle
formation based on a classical nucleation model. They use a model for the hydrate formation relying on *ab initio* calculations
of small sulfuric acid clusters and on experimental data for vapor pressures and equilibrium constants for hydrate formation.
The parameterized formulas are valid at temperatures between 230.15 K and 305.15 K, RH$_w$ from 0.01%-100%, and sulfuric
acid concentrations from 10$^4$-10$^{11}$ cm$^{-3}$. Temperatures along the trajectories ranged between 218 and 252 K and thus were
below the applicable temperatures of this nucleation scheme in 18 out of 32 simulated cases. In these low-temperature cases,
we assume the temperature to be 230.15 K (i.e., we do not extrapolate beyond the bounds of the parameterization). When
sulfuric acid concentration was <10$^4$ molecules cm$^{-3}$, the model assumes a nucleation rate of zero, and it limits the maximum
concentration of gas phase sulfuric acid to 10$^{11}$ molecules cm$^{-3}$.
In the Napari et al. (2002) scheme, referred to here as NAPA, the nucleation rate is parameterized using four variables:
temperature, RH$_w$, H$_2$SO$_4$ concentration, and NH$_3$ mixing ratio. The parameterization is valid for temperatures from 240–300
K, RH$_w$ from 5–95%, sulfuric acid concentrations from 10$^4$–10$^9$ molecules cm$^{-3}$, ammonia mixing ratios from 0.1–100 ppt,
and nucleation rates from 10$^{-5}$–10$^6$ cm$^{-3}$s$^{-1}$. When temperature is <240 K or >300 K (25 out of 32 simulated cases), or RH$_w$ is
outside of the limits stated above, the model assumes the temperature to be 240 or 300 K, and RH$_w$ to be 5 or 95 %, respectively.
When the sulfuric acid concentration is <10$^4$ molecules cm$^{-3}$ the model assumes a nucleation rate of zero, and it limits the
maximum concentration of gas phase sulfuric acid to 10$^9$ molecules cm$^{-3}$.





This parametrization accounts only for hydrate formation and neglects the formation of ammonium bisulfate and its
effect on nucleation rate (Zhang et al., 2010). It overpredicts the effect of ammonia on nucleation when compared with
laboratory measurements (Zhang et al., 2010). Merikanto et al. (2007) showed that nucleation rates based on the NAPA scheme
were biased high, and Lucas and Akimoto (2006) indicated that this scheme in a global model predicted unrealistically high
nucleation rates throughout the troposphere. To address these issues, Westervelt et al. (2013) and Jung et al. (2010) used a
nucleation rate tuning factor of $1 \times 10^{-5}$ in the boundary layer and found that the model produced a reasonable agreement with
observations. In this study we performed simulations both with (NAPAt) and without (NAPA) this tuning factor.
In Dunne et al. (2016), referred to here as DUN, the inorganic nucleation rates determined experimentally in the
CLOUD chamber are parametrized in four dimensions: sulfuric acid, ammonia, temperature (208-292 K) and ion formation
rates (0-75 $cm^{-3}$ $s^{-1}$). Humidity is not included in this parametrization. The overall nucleation rate is given by the sum of the
individual processes

$$J_{b,n} = k_{b,n}(T)[H_2SO_4]^{p_{b,n}} \tag{1}$$

$$J_{t,n} = k_{t,n}(T)f_n([NH_3],[H_2SO_4]) \tag{2}$$

$$J_{b,i} = k_{b,i}(T)n_-[H_2SO_4]^{p_{b,i}} \tag{3}$$

$$J_{t,i} = k_{t,i}(T)n_-f_i([NH_3],[H_2SO_4]) , \tag{4}$$

where $J_{b,n}$ is the binary neutral rate, $J_{b,I}$ is the binary ion-induced rate, $J_{t,n}$ is the ternary neutral rate, $J_{t,i}$ is the ternary ion-
induced rate, $n_-$ is the steady state concentration of small negative ions and $[H_2SO_4]$ and $[NH_3]$ are gas concentrations ($cm^{-3}$).
In this paper, we investigated separately ion-induced binary (DUN with $NH_3$ set to 0) scheme as well as the overall nucleation
scheme (DUN) given by the sum of the above.
Sulfuric acid-organic nucleation was simulated using the scheme described in Riccobono et al. (2014), referred to
here as RIC. While this scheme was developed to represent terrestrial organic species, we use it here as a surrogate for marine
organic compounds because there are no specific mechanisms that have been developed for remote marine-sourced precursors.
The model includes a secondary organics aerosol precursor (SOAP; MW=200 g $mol^{-1}$) variable, which can oxidize
to form a condensable aerosol species. This species can both participate in nucleation in the RIC scheme and condense onto
particles in all schemes studied here. We assume a reaction rate constant for the oxidation of biogenic organic species against
OH is $\sim 3 \times 10^{-12}$ $cm^3$ $s^{-1}$ $molec^{-1}$, which is roughly an average reaction rate of non-methane alkanes according to Table 1 of
Atkinson and Arey (2003). This rate constant gives a SOAP lifetime of ~2 days for a typical diurnally averaged UT OH
concentration of $2 \times 10^6$ $cm^{-3}$. The yield of SOAP to secondary organic aerosol (SOA) is set to 1, which allows us to use SOAP
as a simple, tunable variable to determine how much SOA may be necessary to match observed aerosol formation and growth.
We use the SOAP oxidation product (i.e. condensable organic) in the RIC scheme, but also use it to explore the effects of
organics on new particle growth for each of the nucleation schemes (Riipinen et al., 2011).
In the RIC mechanism, nucleation occurs when only a fraction of the oxidation products of biogenic organic
compounds (*BioOxOrg* in the terminology of the RIC mechanism), formed from SOAP oxidation, are able to form stabilized





clusters. The formation rate dependence on sulfuric acid and *BioOxOrg* concentration is given by a fit to experimental data in
the form

$$J_{ORG} = k_{NUC}[H_2SO_4]^p[BioOxOrg]^q, \tag{5}$$

where $J_{ORG}$ is the formation rate (cm$^{-3}$ s$^{-1}$) of stable particles with diameters ~1.7nm, $k_{NUC}$ is the nucleation rate constant with
a value of 3.27x10$^{-21}$ cm$^6$ s$^{-1}$ at 278 K and RH$_w$ at 39 %, *BioOxOrg* represents concentration of later generation oxidation
products of biogenic monoterpenes (cm$^{-3}$), and the exponents $p=2$ and $q=1$ represent the power law dependence of $J_{ORG}$ upon
the concentrations of sulfuric acid and *BioOxOrg*.

Using the RIC scheme, we test the effect of different fractions of condensable organic formed from SOAP oxidation.

This fraction, $F_{orgnuc}$ represents the fraction of the condensable *BioOxOrg* that may participate in nucleation by stabilizing the
cluster. The value of $F_{orgnuc}$ does not affect the condensation of organics onto already-nucleated or pre-existing particles. Using
$F_{orgnuc}$ allows us to decouple the possible role of organics in nucleation vs. their role in subsequent condensational growth.

Since RIC scheme does not consider the possible effect of temperature on the nucleation rate, we modify the

nucleation rates predicted in equation (5) using the temperature dependence (270-310 K) for this nucleation rate from Yu et
al. (2017)

$$J_{ORG-T} = J_{ORG}f_T \tag{6}$$

$$f_T = exp\left[\frac{\Delta H}{k}\left(\frac{1}{T} - \frac{1}{T_0}\right)\right], \tag{7}$$

where $f_T$ is the nucleation rate scale factor accounting for the temperature dependence, and $\Delta H$ is the change in enthalpy of -
38.3 kcal mol$^{-1}$ associated with the critical cluster formation. We assume that $\Delta H$ is constant throughout our full temperature
range.

One of the limitations of our box modeling effort is that the temperatures along the trajectories ranged between 218

and 252 K, often below the applicable temperatures of the three nucleation schemes: VEHK, NAPA and RIC (Supplemental
material Table S2). We would expect faster nucleation rates at the lower trajectory temperatures than are simulated by these
schemes (e.g. Yue and Hamill, 1979). Using VEHK and NAPA schemes below their lower temperature limit means forcing
them to their lowest rated temperature 230.15 K and 240 K respectively. This in turn may result in underestimating particle
concentration and size. This bias for cold cases means that VEHK and NAPA schemes may predict SO$_2$ and organic precursors
that would be anomalously high. In the RIC scheme the temperature dependence of Yu et al. (2017) is not experimentally
verified down to the tropical UT temperatures. Thus, we tested the impact of changing the $\Delta H$ by ± 3 kcal mol$^{-1}$ (Supplemental
material Fig. S2). We also have not explored the organic-only nucleation scheme by Kirkby et al. (2016).
**2.4 TOMAS input data**

Measured and estimated inputs needed to initialize the TOMAS box model (Adams and Seinfeld, 2002; Pierce and

Adams, 2009; Pierce et al., 2011) are given in Table 1. TOMAS was configured to use measured size distributions (>12 nm)
in discrete bins. Each input in Table 1 represents the initial conditions present at the start of the simulation ($t_0$). Hence,





condensing vapor in the gas phase can contribute both to the formation and growth of new particles and growth of the pre-
existing background aerosol.
We expect the output of the TOMAS model to be sensitive to the temperature dependence of nucleation rates, the
type and number of organic compounds, $SO_2$, OH, $NH_3$ mixing ratios, and the pre-existing background aerosol into which the
convective outflow is mixed. The variability of the simulated aerosol size distribution to various initial conditions was
examined by conducting sensitivity simulations (Table 1) on $SO_2$, $NH_3$, OH, background aerosol size distribution, organics
added for initial growth (e.g., SOAP), and on the RIC scheme scale factor $F_{orgnuc}$ for organics involved in cluster formation.
The pre-existing aerosol is estimated based on the 1-minute averaged size distributions for $D_p$>12 nm as observed at
the aircraft location. The concentration of particles with $D_p$ <12 nm is set to zero under the assumption that these particles were
produced by the new particle formation being modeled and were not present in the background air at the point of mixing with
the air detrained from convection. The box model simulations do not explicitly account for the mixing of highly scavenged air
detraining from convective outflow with surrounding UT air containing more aged aerosol (e.g., Weigel et al. 2011). We have
undertaken sensitivity studies that vary the pre-existing background aerosol used as initial input parameter (Table 1).
The box model simulations were run forward in time from the moment the parcel exited the convection ($t_0$) to the
point of measurement by the aircraft ($t_{fin}$), with temperature, pressure, and $RH_w$ varying as a function of time as determined
from the back trajectory. The concentration of OH at solar zenith angle of 0° in the simulations was set to $3 \times 10^6$ molecules
$cm^{-3}$; however, we also tested OH concentrations of $1 \times 10^6$ and $4.3 \times 10^6$ molecules $cm^{-3}$. These estimates agree well with
aircraft-measured concentrations (Supplemental Material Fig. S3) and with values given in Seinfeld and Pandis (2006). In
TOMAS, OH is parameterized as a function of the cosine of the solar zenith angle, where the night-time OH is $1 \times 10^5$ molecules
$cm^{-3}$. The solar zenith angle is calculated for the time, altitude, latitude, and longitude of the back trajectories.
The $SO_2$ and $NH_3$ mixing ratios were varied between 1 and 100 pptv to explore a large range of plausible conditions.
The evaluated $SO_2$ range exceeds that measured on ATom 4 (Supplemental Material Figs. S4 and S5) and covers the <30 pptv
mixing ratios previously reported in the UT over the central and western tropical Pacific (Thornton et al., 1997; Rollins et al.,
2017; Rollins et al., 2018). Organic aerosol precursors are unknown in the UT and were not directly measured; thus we explored
a range of probable mixing ratios between 1 and 100 pptv.

Table 1. Ranges of parameters used for sensitivity studies in the TOMAS box model. Values varied to match the observed size
distribution in *italic*).

| Parameter | | | Initial value used |
|---|---|---|---|
| Abbreviation | | Unit | TOMAS |
| $SO_2$* | | | *1-100* |
| $NH_3$ | | pptv | *1-100* |
| Secondary organic aerosol precursors (SOAP) | | | *1-100* |
| $F_{orgnuc}$** | | % | *10, 50, 100* |
| OH at solar zenith angle of 0° | | molecule $cm^{-3}$ | *$1 \times 10^6$*, $3 \times 10^6$, *$4.3 \times 10^6$* |





| OH at night | | $1 \times 10^5$ |
|---|---|---|
| Napari et al. (2002) scheme; nucleation rate tuning factor $1 \times 10^{-5}$ | | tuning factor on/off |
| Time since CI | hours | 0.4-23.3 |
| Ion concentration | $cm^{-3}s^{-1}$ | 15 |
| Background pre-existing aerosol | | |
| Size distribution (SD) | | Varied measured initial input size distribution: ***SD>12nm, SD>12nm x2, SD>12nm /2, SD=0, SD>12nm-5nm, SD>8nm, SD as logarithmic fit |

* SO₂ measured on ATom 4 only
** fraction of SOAP participating in nucleation when using Riccobono et al. (2014) in TOMAS.
*** initial background aerosol size distribution was varied: SD>12nm means background SD as described in the text was used to initiate the
model; SD>12nm x2 means background SD multiplied by 2; SD>12nm /2 means SD divided by 2; SD=0 means no background aerosol;
SD>12nm-5nm means SD was shifted by 5 nm to smaller diameters; SD>8nm means measured background SD >8nm was used as initial
SD.

**2.5 Evaluating simulated size distributions**
To determine which sets of parameters allow the models to reproduce the observed size distributions best, we evaluate
every simulation against observations using the normalized mean error (*NME*) statistic of the first four moments (0[th] through
3[rd]) of the size distribution for each model simulation as
$$NME = \frac{\Sigma_{i=0}^{3} \frac{|S_i - O_i|}{O_i}}{4}, \qquad (8)$$
where $S_i$ and $O_i$ are $i$[th] moments of the simulated and observed size distributions, respectively (Hodshire et al., 2018).
The $i$[th] moment is defined as
$$M_i = \int_{2.6}^{20} n_N D_p^i dD_p \ , \qquad (9)$$
where $n_N$ is the number of particles in size interval $dD_p$ and $D_p$ is the diameter. Equation (9) is integrated over the diameter
range from 2.6-20 nm, and $M_i$ represents either $S_i$ or $O_i$. The zeroth moment ($i$=0) corresponds to the number of particles, the
first moment ($i$=1) to the total diameter of particles (i.e. total aerosol length), the second moment ($i$=2) is proportional to the
total surface area of particles, and the third moment ($i$=3) is proportional to the total volume of particles. A *NME* of 0 is a
perfect fit between the simulation and observations; a *NME* of 1.0 indicates that the average bias of the 0[th] through 3[rd] moments
between the simulation and observations is 100%. As the *NME* is given as an absolute value, we do not discriminate between
cases in which the model is underpredicting or overpredicting the moments on average. Since these moments are equally
weighted, a low value of *NME* can be achieved only if the modeled size distribution accurately simulates both the shape and
magnitude of the observed size distribution over the full range of sizes evaluated.





## 3. Results

### 3.1 Observations

Our data show seasonally persistent high nanoparticle concentrations over the remote tropical UT (Fig. 2; Williamson et al., 2019). In this region, the highest concentrations of particles were in the nucleation mode (3-12 nm), which have a short lifetime and are the products of recent NPF. This tropical UT feature was observed in all ATom deployments over all four seasons, over both the Pacific and Atlantic basins. The concentrations of particles observed in the UT over the tropical Atlantic were lower in concentration than observed over the Pacific (Supplemental Material Fig. S6). In this study, we focus on observations over the remote Pacific, which is a region less perturbed by continental emissions than the Atlantic (Fig. 3 and Supplemental Material Fig. S7 and S8).



**Figure 2: Ambient pressure as a function of latitude colored by the measured number concentration of particles with $D_p$ from 3 - 60 nm over the Pacific Ocean for a) ATom 1, July-August 2016; b) ATom 2, January-February 2017; c) ATom 3, September-October 2017; and d) ATom 4, April-May 2018). Periods of flight in clouds, over continents and near airports have been removed.**

Previous studies (e.g. Clarke, 1992, 1993; Clarke et al., 1998, 2006; Brock et al., 1995; Weber et al., 1995; Raes et al., 1997; Thornton et al., 1997; Weber et al., 1998; Clarke and Kapustin, 2002; Twohy et al., 2002; Froyd et al., 2009; Borrmann et al., 2010; Weigel et al., 2011) have provided strong evidence of NPF in the tropical UT and its link to convective activity. However, these earlier studies did not provide such extensive, representative, and global-scale coverage of the remote





marine troposphere over a wide range of altitudes and latitudes (Williamson et al., 2019). The ATom observations also provide
accurate and sensitive, state-of-the-science measurements of the chemical composition of the bulk aerosol and the abundance
of hundreds of gas-phase species in all four seasons, making these observations the most comprehensive to date. However, no
measurements were made during ATom of $NH_3$, the highly oxygenated organic molecules that are likely aerosol precursors,
or molecular cluster composition, and measurements of $SO_2$ took place only during the fourth ATom deployment.

Ten-day back trajectories in the region of NPF in the central Pacific showed transport primarily over the Pacific, with

some possible terrestrial influence from the western Pacific region (Fig. 3). However, trajectories coming from the western
Pacific generally stayed at high altitudes and did not show recent convective uplift from regions influenced by terrestrial
sources. Further, CO and other continental tracers were at background levels over the Pacific, confirming little continental
influence in the sampled air masses (Supplemental Material Fig. S7), as opposed to the Atlantic (Supplemental Material Fig.
S6 and S7). Thus, the precursors of the recently formed particles are likely mostly marine in origin. The latter is also supported
by the measurement of particle phase methanesulfonic acid (MSA) that can be considered as a tracer for maritime influence
on the tropical UT (Fig. S9).




**Figure 3: Flight track and selected 10-day back trajectories initiated for times in flight at pressures <400 hPa (<~260 K) sampled along the DC-8 flight track during ATom 1, 2 (a, b), 3 and 4 (c, d) during the most tropical flight in each deployment (Hawaii-Samoa on ATom 1 and Hawaii-Fiji on ATom 2-4). Trajectories are colored according to the pressure along their pathway.**

The observations and trajectory modeling show that newly formed particles were often associated with deep convection. Using the CI probability criterion of 95% to identify when the aircraft was sampling air recently influenced by convection (Sect. 2.2), and considering the latitude range 30° S - 30° N and ambient temperatures <260 K, for ATom 1 and 2, the shorter the time since convection, the higher the number of small particles (Fig. 4a-d). Such strong trends were not evident for ATom 3 and 4 indicating that factors other than time since CI affect nucleation-mode concentrations. The more recent the convection, the smaller the diameter of the nucleation mode (Fig. 4e-h). These relationships are again strongest for ATom 1 and 2 and also 4. Our hypothesis for these relationships is that with increasing time since CI, particles with diameters <12 nm grew by condensation and coagulation and decreased in concentration by coagulation, leading to the decrease in nucleation-mode concentration and increase in diameter. A similar trend was observed over the Atlantic (Supplemental Material Fig.



S10). The highest concentrations of nucleation-mode particles occurred during ATom 2 and were associated with the shortest
times since CI.


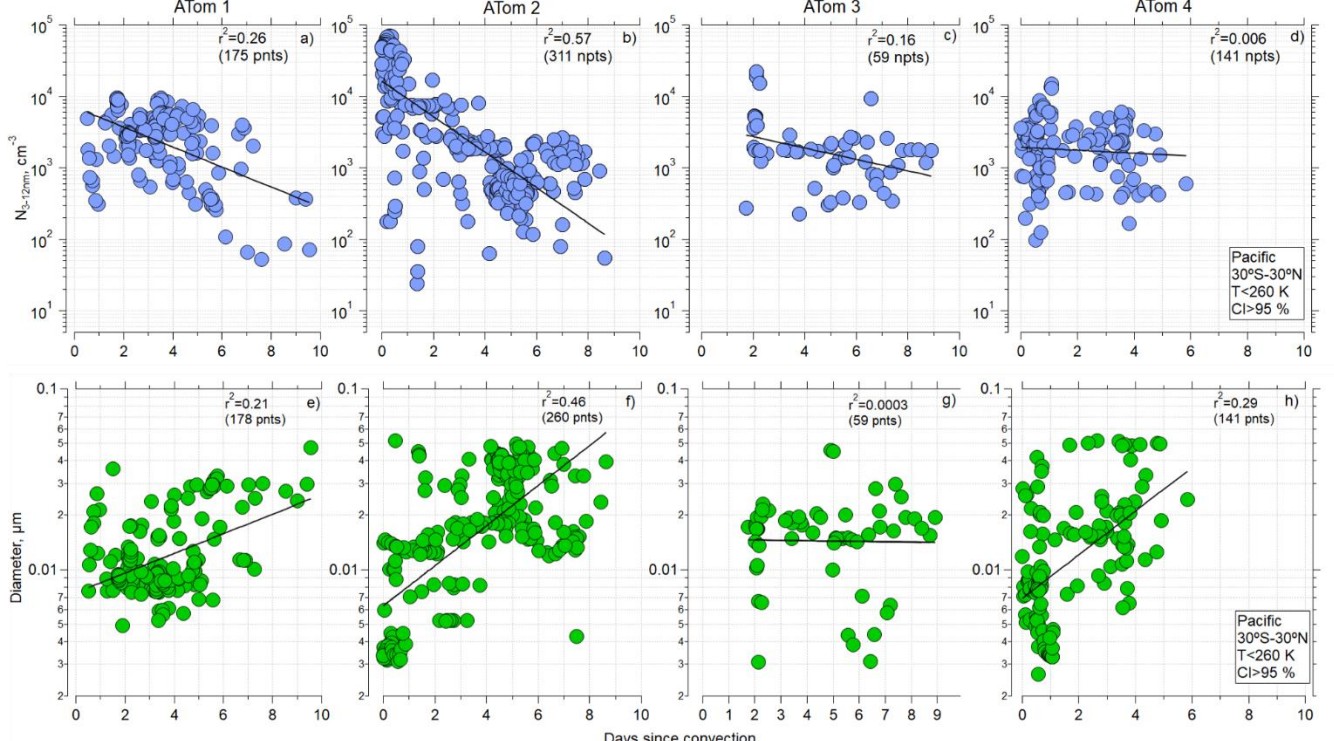


**Figure 4: (a-d) Concentration of nucleation mode particles as a function of time since convective influence for ATom 1-4, over Pacific**
**(30⁰ S - 30⁰ N), T<260 K and probability of convective influence >95 %, respectively. (e-h) Modal diameter of particles with $D_p$<0.06**
**μm as a function of time since convective influence (30⁰ S - 30⁰ N) for ATom 1-4, respectively. Black line, used to guide the eye,**
**represents the linear regression fitted to log-y values. A corresponding Pearson correlation coefficient r² is indicated.**

Air detraining from deep convection is likely depleted in pre-existing particles due to in-cloud removal, leading to a

reduced condensation sink (CS) that enhances the likelihood of NPF (e.g., Clarke, 1992; Williamson et al., 2019). Figure 5
shows the concentration of measured nucleation-mode particles as a function of altitude for the Pacific basin over four ATom
missions. The median concentration of nucleation mode particles averaged from 30° S to 30° N is highest at altitudes >10 km,
reaching ~40,000 cm⁻³ (Fig. 5a), coinciding with the lowest values of CS, which competes with NPF for condensing vapors.
The CS term is calculated for particle diameters >7 nm following Williamson et al. (2019). Over the Altantic, the maximum
concentration of nucleation-mode particles >8 km in altitude averaged from 30° S to 30° N, ~3,000 cm⁻³, is considerably
smaller than over the Pacific, but the shape of the profile is similar (Fig. S11).

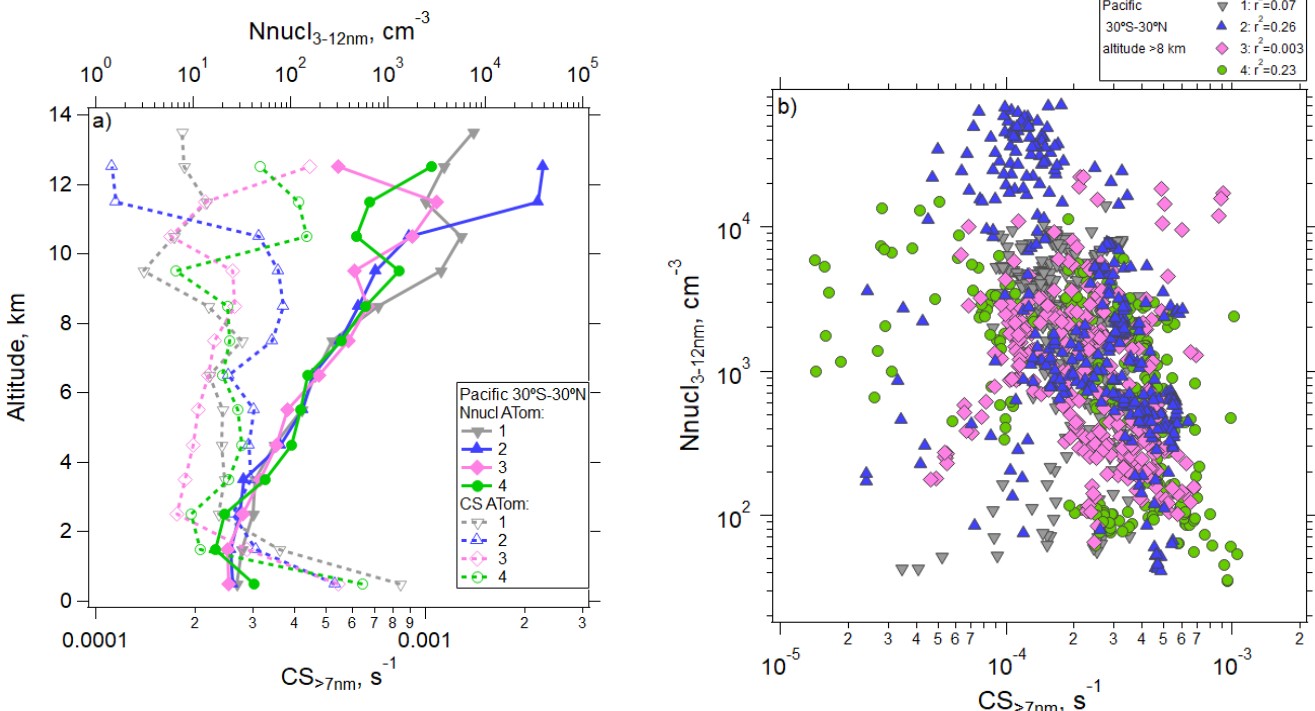

**Figure 5. a)** Vertical profile of the median number concentration of nucleation mode particles (3-12 nm) and condensation sink (CS) averaged between 30° S - 30° N as a function of altitude for the four ATom deployments. **b)** One minute average nucleation mode particle concentrations at >8 km in altitude as a function of CS between $30^0$ S - $30^0$ N over the Pacific Ocean. Pearson correlation coefficient values ($r^2$) are indicated in the legend.

Some variability in the strength of NPF and its dependence on CS can be observed. In general, CS is weakly negatively correlated ($r^2$ between 0.03 and 0.26 depending on the ATom mission with the concentration of nucleation mode particles (Fig. 5b), as would be expected if NPF were competing with CS for condensing vapors. Factors other than CS are also important in controlling the concentrations of newly formed particles. These factors may include temperature and $RH_w$, actinic flux, and OH that drive photochemical reactions that oxidize precursor species, the abundance of those precursor species in the air lifted by convection and in the background air, and the time since the air parcel exited a convective cloud (Figs. S12-S13).

**3.2 Box model simulations**

Case studies were selected for box model simulations based on specific criteria such as temperature and CI probability. We restrict the analysis to data taken nominally in the tropics and subtropics, between 30° S - 30° N latitude. We consider the case for analysis by box modeling if the CI probability is >95%, temperature at the point of measurement is <260 K, and an aerosol number mode with a modal peak diameter <12 nm is present (Table 2). We performed simulations for 32 cases randomly selected from the ATom 2 (20 out of 47 identified cases) and ATom 4 (12 out of 60 identified cases) datasets over the remote tropical Pacific from the total number of 109 cases with time since convection <1 day (Table 2). Data from



ATom 2 and ATom 4 were selected for simulations because these deployments had the most identified cases with time since
CI <1 day. During ATom 2, we observed the highest numbers of nucleation mode particles, lowest condensation sink, and
shortest time since convection (Fig. 5) among all missions. Measurements of $SO_2$ mixing ratio were made only during the
ATom 4 deployment, providing an important constraint for the box model simulations. We did not perform simulations on
ATom 1 and ATom 3 data as there were only 2 and 0 identified cases with time since CI <1 day, respectively (Table 2).
The correlation between nucleation-mode particles and time since CI was strongest in ATom 2 (Fig. 4), while CO
levels, a proxy for continental influence, were the lowest for trajectory times <1 day (Supplemental Material Fig. S7). Although
$SO_2$ was not measured during ATom 2, we expect $SO_2$ in this region in the UT to be <30 pptv based on $SO_2$ levels measured
during ATom 4 and other missions in the Pacific (Supplemental Material Figs. S4, S5).

Table 2. Number of identified cases of recent NPF associated with CI for the Pacific (Atlantic in Table S4) between 30° N and
30° S latitude that meet the following criteria: T<260 K, CI > 95%, and modal peak diameter < 12 nm.

| ATom mission | Number of cases meeting selection criteria | | |
|---|---|---|---|
| | Trajectory age <1 day | Trajectory age 1-2 days | Trajectory age 2-3 days |
| 1 | 2 | 20 | 49 |
| 2 | 47 | 3 | 4 |
| 3 | 0 | 0 | 5 |
| 4 | 60 | 9 | 2 |
| Total | 109 | 32 | 60 |


The size distributions simulated by TOMAS were smoothed to avoid the artificial distortion of the distribution caused
through size-bin emptying (Hodshire et al., 2019; Stevens et al., 1996). The latter and the smoothing technique are described
in Supplemental Material Section S2.
We performed box model simulations on the 32 selected cases using the range of values listed in Table 1. The success
of each model simulation was evaluated using the *NME* described by Eq. 8. As an example using a single case, Fig. 6a shows
the observed and simulated aerosol size distribution with the best *NME* obtained for each of the various nucleation schemes
tested, along with the corresponding mixing ratios of $SO_2$, $NH_3$, or organics. Organics here refer to the SOAP oxidation product
(i.e. condensable organic) that participates in nucleation in the RIC scheme (as *BioOxOrg)*, and in the particle condensational
growth in all schemes. The value of *NME* as a function of the mixing ratios of $SO_2$, $NH_3$, and organics for each nucleation
scheme is also shown (Fig. 6b-j). The summary of each of the 32 simulated cases is presented in Supplemental Material Table
S4 and Figs. S15-S45. The simulations in Fig. 6 used the default OH scheme with a maximum concentration of $3x10^6$ $cm^{-3}$ at
a solar zenith angle of 0° (Supplemental Material Fig. S3). Sensitivity studies for maximum OH values of $1x10^6$, $3x10^6$ and
$4.3x10^6$ $cm^{-3}$ are presented in Supplemental Material Fig. S46-S50.





TOMAS simulations using VEHK scheme substantially underpredict the observed tropical nucleation-mode number

concentration, with resulting poor values of *NME* (Fig. 6). Sensitivity tests that vary the pre-existing initial (background)
aerosol or completely remove background particles do not change the results significantly (Supplemental Material Fig. S51-
S53). Further, we find that changing initial input parameters such as $SO_2$ and OH as indicated in Table 1 do not improve the
*NME* for VEHK scheme (Fig. 6b; Supplemental Material Fig. S46). Adding organics to grow particles nucleated by the VEHK
scheme, while reducing *NME* slightly, does not provide adequate agreement with the observations. Similarly, the ion-assisted
binary nucleation scheme of the MAIA box model does not provide good matches with observations (Supplemental Material
Table S4).

The NAPA scheme, both with (NAPAt) and without (NAPA) the tuning factor, did not significantly reduce the *NME*

values from the VEHK results for this case (Fig. 6c). However, when organics were added to condense on the particles
nucleated by this mechanism, the *NME* was reduced to 0.17. The RIC nucleation scheme, updated by the temperature
dependence of Yu et al. (2017), provides the best *NME* (NME=0.02) for all the schemes investigated (Fig. 6f) for the case
shown in Fig. 6. We explored this mechanism with 6 more sensitivity simulations, including various combinations of initial
$SO_2$ and organic mixing ratios, to see how sensitive *NME* is to small changes of initial precursor vapor mixing ratios. For the
example case presented in Fig. 6a, organic mixing ratios <10 pptv and $SO_2$ mixing ratios <5 pptv were sufficient to produce
size distributions that matched the observations with good fidelity (*NME* =0.02). Varying the scale factor of organics taking
part in nucleation ($F_{orgnuc}$) did not change the results significantly (Supplemental Material Fig. S48).

The most recently developed NPF mechanism, the ion-induced sulfuric acid-water, referred here as DUN with $NH_3$

set to 0, and the sulfuric acid-water-ammonia (DUN) nucleation scheme from Dunne et al. (2016), did not provide the lowest
*NME* values among the schemes tested, although adding organics for initial growth of the nucleated particles improved the fits
(NME=0.04) (Fig. 6i, Supplemental Material Table S4). The addition of organics resulted in best NME values for DUN in 5
out of the 32 cases simulated.

Overall, a reduction in NME when organics are added for initial particle growth was also observed for other schemes

(Supplemental Material Table S4). Out of 32 case studies, we found 6 cases when the NAPAt with organics for growth and
the tuning factor applied gave lower *NME* values than all other schemes. However, 4 out of these 6 cases require $SO_2$ or $NH_3$
mixing ratios >50 pptv that exceed ATom 4 $SO_2$ observations and literature values in the tropical UT for $SO_2$ of <30 pptv (Fig.
S4; (Rollins et al., 2018; Rollins et al., 2017; Thornton et al., 1997) and for $NH_3$ of <10 pptv (Höpfner et al., 2016; Feng and
Penner, 2007; Adams et al., 1999).




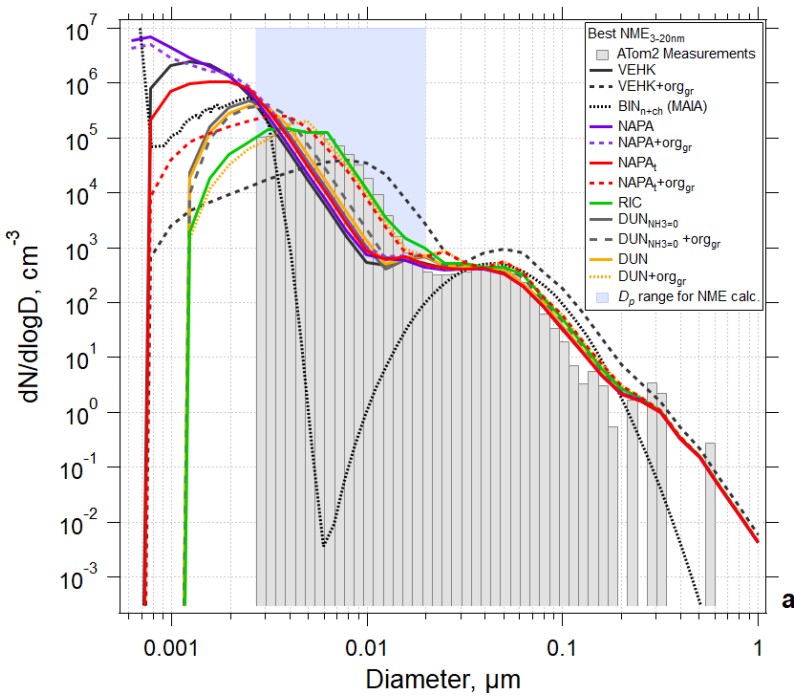

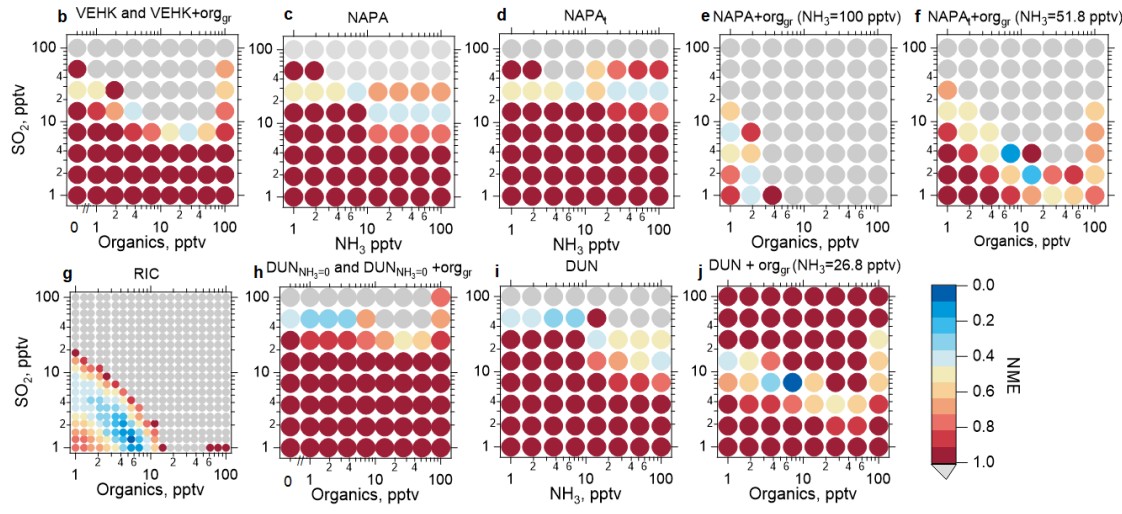

| | VEHK* | VEHK*+org$_{gr}$ | BIN$_{n+ch}$ (MAIA) | RIC* | NAPA* | NAPA*+org$_{gr}$ | NAPA$_t$* | NAPA$_t$*+org$_{gr}$ | DUN$_{NH3=0}$ | DUN$_{NH3=0}$+org$_{gr}$ | DUN | DUN+org$_{gr}$ |
|---|---|---|---|---|---|---|---|---|---|---|---|---|
| NME$_{3-20nm}$ | 0.48 | 0.41 | 0.73 | 0.02 | 0.42 | 0.38 | 0.39 | 0.17 | 0.37 | 0.30 | 0.34 | 0.04 |
| SO$_2$, pptv | 26.8 | 7 | 26.8 | 1.3 | 13.9 | 7.2 | 26.8 | 3.7 | 51.8 | 51.8 | 51.8 | 7.2 |
| Organics, pptv | | 26.8 | | 5.5 | | 1 | | 7.2 | | 1 | | 7.2 |
| NH$_3$, pptv | | | | | 13.9 | 100 | 100 | 51.8 | | | 7.2 | 26.8 |

*Temperature along the trajectory does not lie within the temperature range of the scheme

**Figure 6: Results of simulations using the TOMAS box model for an example case (ATom 2, 2017-02-04, 03:05:31-03:06:30 UTC) where measurements were made 7.3 hours following convective influence, and temperature along the trajectory varied between 218 and 226 K. (a) Observed (shaded bars) and simulated (lines) aerosol size distributions with best normalized mean error (*NME*)**





calculated for $D_p$ between 2.6 and 20 nm (blue shading) for each of the NPF and growth schemes investigated. Best results from the
MAIA box model ion-assisted + neutral binary nucleation scheme shown as a dotted black line. (b) *NME* between the modeled and
measured size distribution for the VEHK scheme with varying organics mixing ratios for condensational growth. The color of the
circle indicates the value of *NME* corresponding to a particular initial mixing ratio of $SO_2$, $NH_3$, or organics that varied between 0
and 100 pptv. Blue represents the best agreement, red poorer agreement, and grey the worst (*NME* >1). There were 64 sensitivity
tests. (c) As in (b), but for the NAPA scheme. d) As in (c), but for the NAPAt scheme. (e) and (f) as in (c) and (d) respectively, but
with $NH_3$ fixed and varying organics for condensation growth. (g) as in (b) but for the RIC scheme, which provides the lowest *NME*.
There were 400 sensitivity tests for this scheme. (h) as in (b) but for the DUN scheme with $NH_3$ set to 0 ($DUN_{NH3=0}$). (i) as in (c) but
for the DUN scheme. (j) as in (i) but with varying organics for condensation growth. The table presents the *NME* results for the
corresponding size distributions in panel (a) and associated initial mixing ratios of gas-phase precursors.

Figure 7 shows the time evolution for particle number concentration, surface area, and volume for the nucleation,

Aitken, and accumulation modes using the Riccobono et al. (2014) scheme for the same case as shown in Fig. 6 for the
simulation with the lowest NME in Fig. 6g ($SO_2$=1.3 pptv, organics=5.5 pptv). There is rapid evolution of the nucleation mode
and slower changes of the larger modes, and the model effectively matches the number, surface and volume of the measured
nucleation mode.

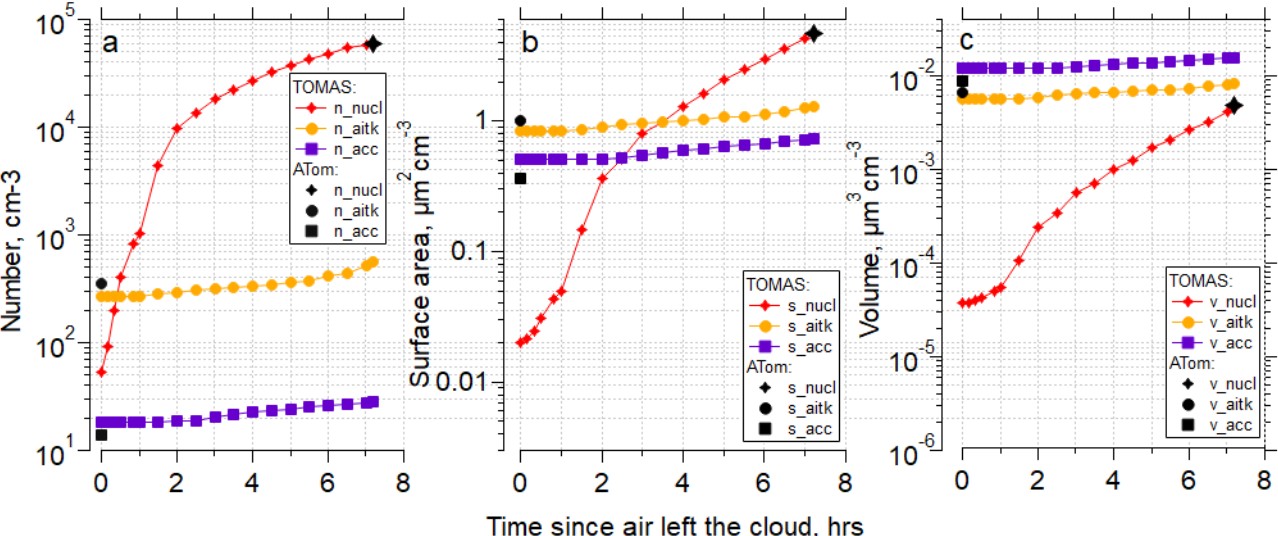


**Figure 7:** (a) TOMAS box model simulation of the case shown in Fig. 6 for the lowest (best) NME for the RIC scheme, showing
number concentration of the nucleation (3-12 nm), Aitken (12-60 nm) and accumulation (60-500 nm) modes as a function of time
since the air parcel exited the cloud to the time of measurement by the aircraft. Black symbols indicate values at the point of
measurement. The measured Aitken and accumulation mode values from the observations were used as approximate initial
conditions for the model simulation and are thus shown at time t=0 (cloud outflow). (c) as in (a), but for surface area concentrations.
(c) as in (a), but for volume concentrations. Conditions for the simulations were diurnally varying OH concentrations with solar
zenith angle. Initial $SO_2$=1.3 pptv, and initial organics=5.5 pptv.






In 22 out of the 32 cases for which multiple box-model simulations were run, the sulfuric-acid-organic nucleation

scheme of (Riccobono et al., 2014) produced lower (better) values of *NME* than the other parameterizations tested
(Supplemental Material Table S4). Two of those 22 best NME cases for RIC were tied with NAPA and NAPAt, both with
organics added for initial particle growth. The remaining 12 best NMEs came from two different ternary nucleation schemes
with added organics for growth of particles. These schemes were the NAPA or NAPAt, or the DUN with both charged and
neutral channels. The majority of these ternary cases, however, required initial conditions of $NH_3$ of 52 pptv or more, much
greater than the mixing ratios expected at these locations in the UT (Höpfner et al., 2016). Regardless of the available $NH_3$,
together these results strongly suggest that pure binary sulfuric acid-water nucleation, whether ion-assisted or neutral, and
whether coupled with organic growth or not, generally cannot explain the ATom observations. While we are limited by the
lack of direct observations of $NH_3$, amines, and condensable organic species, it is plausible that there are enough of these
compounds–a few to tens of pptv–to participate in ternary nucleation and subsequent growth to be consistent with the ATom
measurements.

The findings for the case of organic-mediated NPF are summarized in Fig. 8, where we show the $SO_2$ and organic

precursor mixing ratios for all sensitivity simulations with NME<0.2 for all the cases analyzed using the RIC scheme,
highlighting the assumptions that yielded the lowest NME for each case. The results show that for all of the cases where
sulfuric acid-organic nucleation most successfully simulated the observations (22 of 32 cases), initial $SO_2$ mixing ratios <30
pptv and organic precursors <100 pptv (with an assumed yield of 1) were needed. These $SO_2$ mixing ratios are consistent with
observations during ATom 4 (Figs. S4, S5, Table S5) and earlier results (Rollins et al., 2017; Rollins et al., 2018). Lacking
measurements of condensable organic species, we can only speculate that a few to tens of pptv are reasonable for the marine
tropical UT. Williamson et al. (2019; Extended Data Fig. 7) suggested that organics dominate the composition of smaller
particles at pressure <400 hPa. We note that we performed no simulations with mixing ratios of $SO_2$ or organics above 100
pptv. While we cannot exclude that for some cases the mixing ratios of these precursors at levels above 100 pptv could improve
fits, these levels are outside of prior observations so were not considered in this study.

In the case shown in Fig. 6, mixing ratios of $SO_2$ and organics of ~1.3 pptv and 5.5 pptv, respectively, were sufficient

to nucleate particles and produce a size distribution that matched the observations with an *NME* of 0.02 using the RIC scheme.
In a majority of the cases, the RIC scheme predicted $SO_2$ <5 pptv that are lower than typical UT $SO_2$ concentrations, suggesting
that our temperature extrapolation may overpredict nucleation rates at the typical $SO_2$ mixing ratios of ~30 pptv in the UT.
Overall, the lowest *NME* values were obtained when initial $SO_2$ values were low (<30 pptv), while organics varied over a
range of mixing ratios as shown by triangles in Fig. 8. This suggests that organic matter will often contribute significantly to
the composition of the nucleated and growing particles on a mole basis, and even more so on a mass basis because the assumed
molecular weight of organic precursors and products is 200 g mol$^{-1}$ compared 96 g mole$^{-1}$ for $SO_4$.



**Figure 8: Values of *NME* (colored symbols) for best fits of the sensitivity studies. TOMAS model simulations were made using the**

**RIC sulfuric acid-organic scheme. Among the sensitivity tests using this scheme, the one with the lowest *NME* case is shown with a**

**triangle located at the initial conditions of SO₂ and organics for that case, while the next best *NME* case (provided NME < 0.2) is**

**shown as a circle. The shaded region represents the approximate parameter space in which the best agreement between model and**

**measurement is found for all the convective influence cases studied. Note a different NME color scale range (0 - 0.4) than the one**

**presented in Fig. 6 (0 - 1).**




### 3.3 Discussion


Comparing aerosol size distribution measurements with box-model simulations shows that none of the binary neutral
or ion-assisted NPF schemes are consistent with observations, regardless of precursor concentrations and the presence or
absence of condensing organics for further growth. These schemes predict significant nucleation but do not make enough
particles in the 5-20 size range (Fig.6) to match observations. Adding organics for initial growth of particles shifts the size
distribution to bigger sizes but only slightly improves the model-to-measurement fits (Table S4).
However, schemes that incorporate organic compounds or $NH_3$ to nucleate particles, plus condensing organics as
growth agents, can plausibly replicate the observed size distributions. These results suggest that organic precursor species are
likely important in NPF and initial growth in the tropical upper troposphere, even above marine regions remote from
continental sources. In general, the RIC scheme provided best model-to-measurement fits; however, the improvement in the
fit values for DUN scheme when organics are added for initial growth of particles suggests that organics may be more important
for growth than for nucleation (Table S4).
We find that to best reproduce both nucleation and growth rates by the RIC scheme, the mixing ratios of gas-phase
organic precursors generally needs to be at least twice that of $SO_2$ (Fig.8). While an example in Figure 6 shows that the source
of condensable organics may be even ~5 times the $SO_2$ mixing ratio in the remote tropical UT (Fig. S66), we do not know
whether or not there may be that much more organic precursor available in this region. Although, regions where the oceanic
source of SOA may be higher than the DMS source have been reported previously (e.g. Croft et al., 2019).
Unfortunately, we have no information on the nature and mixing ratios of oxidized organic species that participated
in NPF and initial growth in this environment. The mixing ratios used in this study do not seem out of the range of possibility.
Potential precursors to these condensing species, such as isoprene or monoterpenes (e.g. alpha- or beta-pinene), were found to
be below the limit of instrument detection (2 pptv for isoprene, 0.1 pptv for alpha-pinene, and 0.2 pptv for beta-pinene) in the
tropical UT during the ATom deployments. The exact identification of these condensing organic species would require
instrumentation such as an atmospheric-pressure-interface time-of-flight (API-TOF) mass spectrometer to measure the
composition of molecular clusters, which was not a part of the suite of instrumentation during the ATom mission. Other studies
also suggest that NPF and growth involving organic species may be common in the remote troposphere. Willis et al. (2016)
showed that marine organics contribute to the growth of newly formed particles in the summertime Arctic at low altitude;
however, it was unclear if marine organics were involved in nucleation. Burkart et al. (2017) found that particle growth in the
remote Arctic was largely due to condensation of unidentified organic compounds, possibly of marine origin, associated with
oxidation or photochemistry of the sea-surface micro-layer (Abbatt et al., 2019). Andreae et al. (2018) proposed that oxidized
biogenic VOCs were the source of recently formed particles found in the outflows and anvils of convective storms over
Amazonia.
Chemistry-climate models rarely include organic-driven nucleation pathways in the UT where globally significant
NPF takes place. This may result in poor estimates of NPF and CCN abundance and contribute to uncertainties in aerosol-





cloud-radiation effects. Williamson et al. (2019) showed that the production of newly formed particles and their growth to
cloud-active sizes during descent towards the surface is not adequately captured in the global chemical transport models, which
tend to underestimate the magnitude of tropical UT NPF and subsequent growth. This underestimate might be related to
missing organic precursors, missing chemical mechanisms, or structural errors associated with convective parameterizations.
According to Williamson et al. (2019), the combined direct and indirect radiative effect of NPF in the tropical UT is ~0.1 W
$m^{-2}$, globally.

612   The assumptions in our box model simulations point to the need for further observational and modeling studies. For
example, we do not directly simulate in TOMAS the oxidation of DMS to $SO_2$ and MSA. However, the $SO_2$ mixing ratios
estimated in this study may serve as a proxy for DMS in the modeling in our study, although the timescale for forming $H_2SO_4$
from $SO_2$ will be incorrect. We had measurements of $SO_2$ only during the fourth ATom deployment, and no measurements of
$NH_3$ or highly oxygenated organic molecules that are likely aerosol precursors. Instead, we have constrained the box model
simulations using reasonable lower and upper limits of their mixing ratios based on literature data and in case of $SO_2$, ATom
4 data. Further, nucleation schemes themselves are simply imperfect parameterizations extrapolated from laboratory
observations. One of the limitations of our box modeling effort is that the temperatures along the trajectories were often below
the lower range limit of three (out of four) nucleation schemes evaluated (Table S5). In these cases (marked with a "*" in
Supplemental Material Figs. 15-45 and Table S5) the best-fit $SO_2$ and organic concentrations are expected to be biased high.
Although experimentally unverified, we incorporated temperature dependence into the Riccobono et al. (2014) scheme after
Yu et al. (2017). We would expect faster nucleation rates at the lower trajectory temperatures than are simulated by these
schemes (e.g. Yue and Hamill, 1979). Further, we have tested the Napari et al (2002) scheme both with and without a tuning
factor of $10^{-5}$ that was developed for continental regions (Jung et al., 2010; Westervelt et al., 2013), an obvious source of
uncertainty when simulating NPF in the UT over the oceans. These are schemes that many models use and they do not appear
to (often) work for this region, possibly due to their limited range of operating temperatures. We also have not explored the
organic-only nucleation scheme described by Kirbky et al. (2016). Further, we did not account for mixing with surrounding
air on the path between the cloud outflow and the point of measurement when running simulations.

630   The limitations described above are important and point out the need to undertake further in situ measurements and
modeling studies to confirm the suspected role of organics in UT NPF and subsequent growth in the remote troposphere. Better
understanding of NPF in the remote UT, and the growth of these particles to cloud-active sizes, could substantially improve
model simulations of the preindustrial atmosphere, would allow better evaluations of the effect of current anthropogenic
perturbations, and could allow more confident predictions of the evolution of the climate and its response to future emission
scenarios. Modeling efforts should focus on developing new nucleation mechanisms based on chamber studies conducted at
temperatures more representative of the UT. Further airborne research should focus on measuring the composition of molecular
clusters, sulfuric acid, organics, and $NH_3$ over the oceans and tropical continental areas. The planned Chemistry of the
Atmosphere: Field Experiment in Brazil (CAFE-Brazil) study is the first expected to combine airborne measurements of





nucleation-mode particle size distributions with API-TOF mass spectrometer measurements of the composition of nucleating
clusters.

**4. Summary**

Airborne observations during the ATom mission show a globally significant source region of newly formed particles
in the tropical and subtropical UT that persists over both the Atlantic and Pacific Ocean basins over all seasons. These particles
are often associated with the outflow from deep convection. Averaged across the tropics and subtropics over the Pacific, the
particle number concentrations were a maximum (reaching as high as ~40,000 cm$^{-3}$) at altitudes above 10 km where the
condensation sink from pre-existing aerosol particles was a minimum. Using back-trajectories to identify convectively
influenced air parcels, the highest concentrations of recently formed particles were generally found where the CI was most
recent, particularly during the first and second ATom deployments. The number concentration of nucleation-mode particles
decreased with time since CI due to the effects of coagulation and condensational growth. During ATom 1 and 2, particle size
increased with time since CI, showing clear evidence for this growth.
We simulated particle nucleation and growth using two box models constrained to follow the calculated trajectories
from the point of convective detrainment to the point of measurement by the aircraft, and we performed sensitivity tests varying
the nucleation mechanisms and initial conditions such as precursor (SO$_2$, NH$_3$, organics), OH, and pre-existing particle
concentrations.
These simulations indicate that nucleation schemes commonly used in global models, such as binary homogeneous
H$_2$SO$_4$ (both neutral by Vehkamäki et al. (2002) or ternary H$_2$SO$_4$+ NH$_3$ (neutral with and without a tuning factor by Napari
et al (2002) and Jung et al. (2010))), as well as the recently developed neutral and ion-induced binary and ternary nucleation
scheme by Dunne et al. (2016), were all inconsistent with observed size distributions in all simulated cases when no organics
were included for growth. This result also held for the binary nucleation mechanisms even when organics were added as a
condensing, but not nucleating, species. Adding organics for initial growth of particles formed by either of tested ternary
schemes (Napari et al. (2002) or Dunne et al. (2016)) provided the best fits in 12 out of 32 simulated cases (Table S4). However,
the majority of these ternary inorganic simulations required initial conditions of NH$_3$ >50 pptv, which is substantially greater
than expected at these locations in the UT (Höpfner et al., 2016).
In contrast, a scheme involving oxidation products of biogenic organics and H$_2$SO$_4$ (Riccobono et al., 2014) gave
results that were most consistent among the various models with observations in 22 out of 32 cases, while in 2 cases it was
tied for the lowest NME with other schemes. These results strongly suggest that organics are involved in NPF and subsequent
initial growth in the remote tropical UT. This supports the finding by Simon et al (2019) that organics, despite their lower
oxidation level and yield at low temperatures, may be important for nucleation and growth in the UT. However, the predicted
SO$_2$ concentrations were often anomalously low (<5 pptv), suggesting that our temperature extrapolation may overestimate





the nucleation rates. While the Riccobono scheme was most consistent, the analysis suggests that multiple nucleation
mechanisms may be plausible across the 32 cases.
We have assumed that the Riccobono et al. (2014) scheme, which was developed from laboratory measurements of
nucleation from the oxidation products of terrestrial biogenic VOCs, represents processes in the remote marine UT of the
tropical Pacific. In fact, there is virtually no information on the nature of oxidized organic species (or ammonia and amines)
that may participate in NPF in this environment. Also, the Riccobono scheme required a large extrapolation in temperature to
simulate UT conditions. Given that NPF in the tropical UT is a major source of CCN over a large portion of the globe
(Williamson et al., 2019), we recommend that future work investigate the species contributing to NPF and growth explicitly,
including direct measurements when possible. Additionally, we recommend studies that focus on potential tropical marine
sources of aerosol precursor gases, the efficiency of their transport to the UT, the products of their oxidation, and the
mechanisms of NPF at temperatures <230 K.
**Data availability**
The full ATom dataset is available as given in Wofsy et al. (2018), and may also be accessed at
https://espoarchive.nasa.gov/archive/browse/atom. Data presented in this analysis are available at the Oak Ridge National
Laboratory (ORNL) Distributed Active Archive Center, (DAAC) Kupc et al. (2020).
**Author contributions**
AK, CW, CB, KF, MD, BW, TB, AR collected data. AK, JP, CB, and AH conceived and designed the study. AK performed
the analysis and wrote the manuscript with help from CB and JP, with contributions from all co-authors. JP, AH, JK provided
TOMAS and MAIA box models and helped with model upgrades. AK performed all model simulations. MD and BW analysed
cloud properties. ER calculated the air parcel back-trajectories.
**Competing interests**
The authors declare that they have no conflicts of interest.
**Disclaimer**
The contents do not necessarily represent the official views of the University of Colorado, the University of Vienna, NOAA
or of the respective granting agencies. The use or mention of commercial products or services does not represent an
endorsement by the authors or by any agency.



## Acknowledgements

We thank Ken Aikin for contributions to this analysis, and the ATom science team and NASA DC-8 flight crew for their contributions to the ATom data. We are grateful for the hard work of the ATom leadership and logistics teams. We thank the Whole Air Sampler (WAS; UCI) and the Trace Organic Gas Analyzer (TOGA; NCAR) teams for access to their data.

## Financial Statement

This work was funded by NASA's Earth System Science Pathfinder Program under award NNH15AB12I and by NOAA's Health of the Atmosphere and Atmospheric Chemistry, Carbon Cycle, and Climate Programs. Agnieszka Kupc was supported by the Austrian Science Fund FWF's Erwin Schrodinger Fellowship J-3613. Bernadett Weinzierl and Maximilian Dollner were supported by European Research Council (ERC) under the European Union's Horizon 2020 research and innovation framework program under grant 640458 (A-LIFE) and by the University of Vienna. Jeffrey Pierce and Anna Hodshire were supported by the US Department of Energy's Atmospheric System Research, an Office of Science, Office of Biological and Environmental Research program, under grant DE-SC0019000; and the NOAA, Office of Science, Office of Atmospheric Chemistry, Carbon Cycle, and Climate Program, under cooperative agreement award NA17OAR430001.

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
