# Peer review of "The potential role of organics in new particle formation and initial"

_Atmospheric Chemistry and Physics, 2020_

## Referee Comment (RC1) · Anonymous Referee #1 · 7 Aug 2020

**Review of "The potential role of organics in new particle formation and initial growth in the remote tropical upper troposphere" by Kupc et al**

The authors present box model studies of new particle formation representative of the remote Pacific upper troposphere sampled during ATom flights. They find the nucleation mechanism of Riccobono et al (2014) represents observations reasonably well. While not well constrained by measurements, organic vapor concentrations had to be at least twice $SO_2$ concentrations to reproduce observed NPF and growth rates. Studies like this will be of interest to the community, and are timely relative to the ATom measurements.

The paper is well structured and well-written and is clearly within the scope of ACP. I recommend publication if the authors can address the following comments.

**Minor comments**

**One very long comment about condensation sink.**
    Nucleation varies very non-linearly with condensation sink, as highlighted very nicely in several papers by Pierce et al among others. I think it is necessary to emphasise the implications of the variability in CS more explicitly. The measured condensation sink (CS) varies by a lot – more than an order of magnitude. The generally poor correlation of the CS to nucleation-mode particle concentrations is explained in part because the CS probably varies significantly along the trajectories, and the CS at the end of the trajectory is not the same as the CS where nucleation happened. Indeed, if I interpret Figure 7b correctly, it appears that nucleation-mode particles dominate the sink in the case shown. Therefore, it's unclear whether the correlation of observed number concentration to CS should be positive or negative. So I am not sure if Figure 5b is the most helpful way to present the data. Could you try splitting the data in Figure 5b into cases where CS is dominated by 7-12nm particles and cases where it isn't? Maybe when the CS is dominated by small particles it will be positively correlated to number concentration and when not, negatively? I realise that if the CS is always dominated by 7-12nm particles, then this won't be useful.
    As another way of digging deeper into this, would it be useful to plot the simulated CS at the time along the trajectory when the simulated nucleation rate is maximum, against the simulated number concentration at the end of the trajectory, just to see if there is a stronger negative correlation? Then maybe compare that to the simulated CS at the end of the trajectory plotted against the simulated number concentration at the end of the trajectory?
    I found your Figure 7 very helpful as it shows when nucleation happens along the trajectory. If vapor concentrations and temperature were prescribed to their average values, nucleation would always happen at the start of the trajectory, as nucleation can only ever increase the condensation sink so the most likely time for it to happen is at the beginning of the simulation – correct? However since temperature and $H_2SO_4$ concentrations vary along the trajectory, this doesn't need to be true. So is figure 7 typical? Do you sometimes see nucleation only a few hours after the start of the simulation? Is the CS/surface area always dominated by nucleation-mode particles?
    At line 470-477, you say that variation in the background aerosol doesn't help the Vehkamaki et al nucleation scheme, which is the scheme that is best constrained by your measurements. But in Figure S52 I see some blue dots. Perhaps (if I'm not wrong) you could say that if the CS were smaller by some factor, the Vehkamaki scheme could explain the observations, or could get reasonably close (NME=0.14), as shown in Figure S52? A NME of

0.14 for Vehkamaki might be just as 'good' as a lower NME for the schemes that include organics and NH3, since these vapors are not constrained by your measurements. Can you consider promoting this figure, and perhaps S51 for the RIC scheme, to the main text, as well as improving the caption (to refer to Table 1 where the various abbreviations are defined) and including a more extensive discussion?

**Other minor comments**
A mention of the update to the Vehkamaki (2002) nucleation scheme by Maattanen et al (JGR, 2017; https://agupubs.onlinelibrary.wiley.com/doi/full/10.1002/2017JD027429) is warranted. The update is valid over a wider range of temperature and humidity, which is relevant here. The new model is also validated against CLOUD measurements, includes ion-induced nucleation, and updated quantum chemistry calculations. Similarly, the ternary H2SO4-NH3 scheme of Yu et al, GMD 2020, has look-up tables ready for a model implementation: https://gmd.copernicus.org/articles/13/2663/2020/gmd-13-2663-2020.pdf . I think there is no need to update the schemes used for this paper, but perhaps they could be mentioned as worth exploring in future work?

What motivates the night-time OH value of $1 \times 10^5$? Production from ozonolysis of terpenes? It seems like quite a high value to me.

It is stated that trajectories generally stayed at high altitudes (line 387). How does the trajectory model handle deep convection? Presumably it is parameterized? Your Figure 3 shows some evidence that some trajectories do originate in the boundary layer, which seems reasonable, but are there not some big uncertainties here even with the ensemble approach adopted? It would be worth referencing and perhaps discussing Andreae et al (2018) (the paper in ACP, already cited) in this context, as they do a similar trajectory analysis in their paper.

Line 424 Atlantic is misspelt.

How does RH vary with time since convection? I'm guessing it doesn't vary very much and so is not super-important, but would be good to confirm this. I note it is represented in your simulations in any case (line 324)

Supplement S2, line 457, and Figure 6 : Could the rather strange size-bin emptying behaviour be avoided by shortening the timesteps of the box model?
Also, is size bin emptying actually responsible for the rather strange kinks in the size distribution frequently observed around 10nm, and which correspond to a much smaller kink in the observed size distributions? In Figure S25, this is especially apparent. The gaps in the size distribution are far larger than a single bin. Do you have some ideas for what might be happening here?

L618: "Nucleation schemes are simply imperfect parameterizations extrapolated from laboratory observations": this is a bit too much generalization; it's not quite true for Vehkamaki (2002) for example. On the other hand, the RIC scheme you prefer to use is one of the more uncertain parameterizations from the CLOUD experiment, because the organic molecules thought to participate in nucleation were not measured directly in the chamber when that parameterization was developed. For example, in the conclusions, perhaps it's worth pointing out that it is not surprising that one of the least well constrained nucleation mechanisms – the RIC scheme, which

is the one you can tune most easily by changing the organic concentration – can be made to agree with observations the best.

L1037 several references are merged together.

---

## Referee Comment (RC2) · Anonymous Referee #2 · 11 Aug 2020

This manuscript tackles a globally important research topic: formation of new aerosol particles in the tropical upper troposphere. The paper is essentially a sensitivity study, aiming to give new insight into which nucleation mechanisms and aerosol precursors, coupled with the initial growth of newly-formed particles, best explain the observed ultrafine particle number size distribution. The conducted analysis is based on box model simulations and statistical analyses of the simulation results. The paper is scientifically sound and relatively well structure. I do have, however, a few issues that should be addressed before the paper is ready for publication.

My major criticism is related to the treatment of aerosol processes in convective outflow

regions. As the authors state, they simulate outflow regions of deep convective clouds using a box model. This is fine as long as both nucleation and growth occur well beyond the region abound cloud boundaries where most of the mixing between cloud outflow and upper troposphere air take place. This may not be the case, as it is quite possible that nucleation and early particle growth take place in the mixing region, or event inside the convective clouds. This would seriously bias the results obtained in the paper. I understand that including the cloud and its immediate outflow region in a box model is almost impossible, and therefor outside the scope of the current paper. However, the authors should bring up this issue more honestly as done in the present paper (brief mentioning on lines 628-629). Furthermore, there are a number of both modeling and observations studied conducted on new particle formation in cloud outflow regions. The authors should better acknowledge such studies when discussion their results, their implications and the associated uncertainties.

Other, minor issues:

The right parenthesis is missing from line 432.

The text on lines 538-543 is not logical. When discusses nucleation mechanism not involving NH3 at all, it is incorrect to say "regardless of NH3 oncentraions", as the outcome of such mechanisms does not depend on NH3 concentrations. Please correct.

The statement on lines 546-548 sounds a bit strange. Is really so that the concentration of both SO2 and organic precursors need to be smaller than some upper limit values to reproduce the observations? Please check out this statement and modify if needed.

---

## Referee Comment (RC3) · Anonymous Referee #3 · 17 Aug 2020

The paper investigates the origin of nucleation mode particles in the upper tropical troposphere and makes use of the extensive ATom dataset in the process. The authors use nucleation and growth box models evaluated along the trajectories of air masses reaching from the outflow of tropical convection until the air was encountered during ATom flights. They find that the nucleation parametrization of Riccobono et al. 2014 can describe the measured datasets best, which indicates organic involvement in nucleation. Also, if growth driven by organics is included in the model, the agreement with observations is enhanced as well, further pointing to the importance of organics in the upper tropical troposhpere.

The paper offers very interesting insights into the important, but notoriously hard to observe topic of upper tropospheric new particle formation. The authors corroborate their conclusions with an extensive set of measurement and model data that are nicely presented. The paper is well within the scope of ACP; it is well written and clearly structured. The methods used are described clearly as well as their limitations. I recommend publication in ACP and only have some minor comments to add:

The paper focusses on nucleation rates, which is very insightful, however, in the model also growth, especially driven by organics is incorporated. You state that the inclusion of organic growth enhances the agreement between model and observations in many cases. However, you do not give a range for the growth rates needed to do that. Can you add a figure and/or short discussion that indicate the range of growth rates used in your model?

Are you in Table 1 discussing the ion concentration as indicated by the given parameter name or the ion pair production rate as indicated by the given unit (cm-3s-1)? Please add a short description in section 2.4 that describes the origin of this quantity. The value 15 in Table 1 would correspond to the ion pair production rate in cm-3s-1 given in Dunne et al. 2016 for the upper tropical troposphere.

l. 322: You state "We have undertaken sensitivity studies that vary the pre-existing background aerosol used as input parameter (Table 1)", but in Table 1 there is no information on how this quantity was varied, as it is only named. Can you give more detail in Table 1 on how you varied the pre-existing aerosol and/or refer to the SI part where you discuss this in more depth?

l. 485: You state that "varying the scale factor for organics taking part in nucleation (Forgnuc) did not change results significantly". Can you add a short discussion on why this is the case? You span one order of magnitude in [BioOxOrg] with the scaling factor values you use, so according to the RIC scheme, this should vary the nucleation rate as well by an order of magnitude. So in Fig S48 I would guess that you would need higher

organics in the F = 0.1 case to match observations than in the F = 1 case. However, the blue dots are practically identical in all cases. So what is the compensating effect for that?

Figure 7: It is a bit misleading that you place the black symbols for Aitken and accumulation mode at x = 0, as these originated from the measurement as you write and the measurement was not taken at x = 0. Even if you use them as starting point for the model, I would still place them at the point of actual measurement.

Figure 7: I would suggest to remove grid lines in between the panels.

l. 577: "5-20 size range": You mean nm?

Figure 2: Add an altitude axis on the right, as in Fig 2 in Williamson et al. 2019

---

## Author Comment (AC1) · 27 Oct 2020

////////////////////////////////////////////////////////////////////////////////////////////////////////////////////////////////////////////////////////////////////

**Response to the Anonymous Referee #1**

The authors thank the reviewer for helpful comments that have improved the manuscript. The discussion below contains the original text by the reviewer and our responses (in blue) along with changes made to the revised manuscript. Additional minor changes to the manuscript are indicated at the end of this document.

**Review of "The potential role of organics in new particle formation and initial growth in the remote tropical upper troposphere" by Kupc et al**
The authors present box model studies of new particle formation representative of the remote Pacific upper troposphere sampled during ATom flights. They find the nucleation mechanism of Riccobono et al (2014) represents observations reasonably well. While not well constrained by measurements, organic vapor concentrations had to be at least twice SO2 concentrations to reproduce observed NPF and growth rates. Studies like this will be of interest to the community, and are timely relative to the ATom measurements. The paper is well structured and well-written and is clearly within the scope of ACP. I recommend publication if the authors can address the following comments.

**Minor comments**
**One very long comment about condensation sink.**
Nucleation varies very non-linearly with condensation sink, as highlighted very nicely in several papers by Pierce et al among others. I think it is necessary to emphasize the implications of the variability in CS more explicitly. The measured condensation sink (CS) varies by a lot more than an order of magnitude. The generally poor correlation of the CS to nucleation-mode particle concentrations is explained in part because the CS probably varies significantly along the trajectories, and the CS at the end of the trajectory is not the same as the CS where nucleation happened. Indeed, if I interpret Figure 7b correctly, it appears that nucleation-mode particles dominate the sink in the case shown. Therefore, it's unclear whether the correlation of observed number concentration to CS should be positive or negative. So I am not sure if Figure 5b is the most helpful way to present the data. Could you try splitting the data in Figure 5b into cases where CS is dominated by 7-12nm particles and cases where it isn't? Maybe when the CS is dominated by small particles it will be positively correlated to number concentration and when not, negatively? I realise that if the CS is always dominated by 7-12nm particles, then this won't be useful.

These are really good points. The reviewer is right that from Figure 7 both measured and simulated nucleation mode dominated the CS in that specific case, which could lead to a positive correlation between nucleation mode particle number concentration and CS. To clarify that we have remade Figure 5a and 5b to show the CS for particles >12 nm (instead of CS> 7 nm; updated plot is shown below) as a better estimate of the CS prior to nucleation starting. The correlation remained negative and the value of the correlation coefficient, $r^2$, increased as compared to the initially presented

CS>7 nm data. When CS is dominated by small particles ($CS_{3-12nm}$) the correlation is strongly positive ($r^2$ between 0.97 and 1, Fig. 5c).

One of the assumptions we have made in simulations is that there is no nucleation mode at the start of the trajectory (at the exit of the cloud). Thus, we eliminated all particles with diameters < 12 nm to initiate the model. Figure 5 below is shown in the updated version of the manuscript.

[Figure]

Figure 5. a) Vertical profile of the median number concentration of nucleation mode particles (3-12 nm) and condensation sink ($CS_{>12nm}$) averaged between 30° S - 30° N as a function of altitude for the four ATom deployments. (b-c) One minute average nucleation mode particle concentrations at >8 km in altitude as a function of $CS_{>12nm}$ (b), and $CS_{3-12nm}$ (c), and between 30° S - 30° N over the Pacific Ocean. Pearson correlation coefficient values ($r^2$) are indicated in the legend.

The following text was updated (line 426; changes to the existing text indicated in italic)

"The CS term is calculated for particle diameters *>12 nm, and for the diameter range between 3 and 12 nm* following Williamson et al. (2019). *$CS_{>12nm}$ serves as an estimate of the condensation sink prior nucleation starting, and it is negatively correlated with the number of nucleation mode particles (Fig. 5b), while the nucleation mode is positively correlated with $CS_{3-12nm}$ (Fig. 5c).* Over the Atlantic, the maximum concentration of nucleation-mode particles >8 km in altitude averaged from 30° S to 30° N, ~3,000 cm$^{-3}$, is considerably smaller than over the Pacific, but the shape of the profile is similar (Fig. S11)."

Line 436: "Some variability in the strength of NPF and its dependence on $CS_{>12nm}$ can be observed. In general, $CS_{>12nm}$ is weakly negatively correlated (*$r^2$ between 0.08 and 0.36* depending on the ATom mission) with the concentration of nucleation mode particles (Fig. 5b), as would be expected if NPF were competing with CS for condensing vapors. *When CS is dominated by small particles ($CS_{3-12nm}$) the correlation is strongly positive ($r^2$ between 0.97-0.1, Fig. 5c).*"

We updated the Figure S11 for the Atlantic case in the Supplemental Material (SM) accordingly.

[Figure]

Figure S11: Vertical profile of the median number concentration of nucleation mode particles (3-12 nm) and condensation sink (CS>12 nm) averaged between 30° S - 30° N as a function of altitude for the four ATom deployments. (b-c) One minute average nucleation mode particle concentrations at >8 km in altitude as a function of $CS_{>12nm}$ (b) and $CS_{3-12nm}$ (c) and between 30⁰ S - 30⁰ N over the Pacific Ocean. Pearson correlation coefficient values ($r^2$) are indicated in the legend.

We also checked whether there were instances when $CS_{7-12nm}$ dominated over $CS_{>12nm}$ and found out there were only 7 instances where $CS_{7-12nm}$ exceeded $CS_{>12nm}$ values.

As another way of digging deeper into this, would it be useful to plot the simulated CS at the time along the trajectory when the simulated nucleation rate is maximum, against the simulated number concentration at the end of the trajectory, just to see if there is a stronger negative correlation? Then maybe compare that to the simulated CS at the end of the trajectory plotted against the simulated number concentration at the end of the trajectory?

We chose to follow the first of these two alternate suggestions by the reviewer.

I found your Figure 7 very helpful as it shows when nucleation happens along the trajectory. If vapor concentrations and temperature were prescribed to their average values, nucleation would always happen at the start of the trajectory, as nucleation can only ever increase the condensation sink so the most likely time for it to happen is at the beginning of the simulation – correct?

Yes, that is correct.

However, since temperature and H2SO4 concentrations vary along the trajectory, this doesn't need to be true.

Yes, and the reason for this is because the OH is not necessarily at its highest concentration at the start, which, along with the condensation sink, determines the pseudo-steady-state concentrations of $H_2SO_4$ and condensable organics.

So is figure 7 typical? Do you sometimes see nucleation only a few hours after the start of the simulation? Is the CS/surface area always dominated by nucleation-mode particles?

In all (32) cases, nucleation starts early in the simulation (within the first 20 minutes). The nucleation, however, can have different strengths. In some cases, although nucleation starts early, the maximum nucleation rates are achieved later as the nucleation rate may increase with time. Some nucleation events are very weak, with $dN/dlog_{10}D_p$ never exceeding 10 cm$^{-3}$ in any of the size bins smaller than 12 nm.

In Table S6 below we show a summary of the measured aerosol surface areas for each of the studied cases. The highest surface area values for each case are shaded. These values are split between nucleation, Aitken, and accumulation modes indicating that the case shown in Figure 7, while not representative of all cases used in simulations, is typical. Table S6 is now added to the SM. A few examples of cases where the value of aerosol surface area was either biggest for Aitken or accumulation mode are presented in Fig. S68-70 below and in the SM.

The following text was added to the revised manuscript: Line 531: "In the case presented in Figure 7b, the surface area measured on ATom is dominated by the nucleation mode particles; however, although frequent, this is not a typical pattern among all other cases studied here. In general, in the cases with the highest surface area, values are split between nucleation, Aitken and accumulation modes almost equally (Table S5, Fig. S68-70)."

Table S5. Summary of the measured aerosol surface area for each of the studied cases. The highest surface area values for each case are shaded.

| ATom2 Case identifier | Surface area, µm² cm⁻³ | | | |
| --- | --- | --- | --- | --- |
| | Nucleation mode 3-12nm | 7-12nm | Aitken mode 12-60nm | Accumulation mode >60nm |
| sd486 (Fig.7 in the manuscript) | 4.71 | 1.90 | 1.00 | 0.37 |
| sd390 | 2.89 | 0.89 | 1.87 | 0.81 |
| sd391 | 2.74 | 0.64 | 1.54 | 1.36 |
| sd400 | 1.14 | 0.25 | 1.88 | 1.44 |
| sd404 | 2.47 | 0.76 | 1.98 | 0.84 |
| sd446 | 2.18 | 0.91 | 3.44 | 1.66 |
| sd448 | 2.05 | 0.68 | 2.95 | 1.17 |
| sd452 | 2.63 | 0.90 | 2.91 | 1.20 |
| sd461 | 1.37 | 0.36 | 2.39 | 0.95 |
| sd470 | 0.95 | 0.47 | 3.35 | 1.01 |
| sd477 | 3.33 | 1.21 | 2.04 | 0.59 |
| sd491 | 2.79 | 0.48 | 1.29 | 0.74 |
| sd496 | 4.22 | 1.32 | 1.79 | 0.75 |
| sd498 | 3.05 | 0.82 | 2.13 | 0.80 |
| sd530 | 0.89 | 0.37 | 1.95 | 3.32 |
| sd533 | 2.13 | 0.95 | 1.88 | 3.33 |
| sd535 | 2.23 | 1.10 | 1.61 | 2.49 |
| sd537 | 2.25 | 0.95 | 1.40 | 0.88 |
| sd540 | 0.53 | 0.07 | 0.90 | 0.63 |
| sd543 | 2.71 | 1.43 | 1.99 | 1.62 |
| **ATom4 cases** | | | | |
| sd22 | 0.44 | 0.33 | 1.05 | 1.41 |
| sd32 | 0.50 | 0.37 | 0.90 | 1.12 |
| sd72 | 0.37 | 0.31 | 1.40 | 1.75 |
| sd75 | 0.86 | 0.62 | 1.11 | 1.36 |
| sd78 | 0.31 | 0.22 | 0.94 | 1.56 |
| sd82 | 0.37 | 0.31 | 1.69 | 2.23 |
| sd132 | 0.09 | 0.06 | 1.13 | 1.76 |
| sd134 | 0.14 | 0.09 | 1.94 | 2.03 |
| sd179 | 0.84 | 0.56 | 0.38 | 0.28 |
| sd183 | 0.17 | 0.08 | 0.52 | 0.40 |
| sd186 | 0.05 | 0.02 | 0.445 | 0.436 |
| sd188 | 0.07 | 0.02 | 0.26 | 0.22 |
| TOTAL number of cases with highest surface area in each size range | 11 | 0 | 10 | 11 |

[Figure]

Figure S68: as in Fig. 7 but for case 461 (NME=0.08). Measured on ATom aerosol surface area is dominated by the Aitken mode (b). Initial $SO_2$=1.6 pptv, and initial organics=23.4 pptv.

[Figure]

Figure S69: as in Fig. 7 above but for case 470 (NME=0.28). Measured on ATom aerosol surface area is dominated by the Aitken mode (b). Initial $SO_2$=1 pptv, and initial organics=3.4 pptv.

[Figure]

Figure S70: as in Fig. 7 but for case 530 (NME=0.13). Aerosol surface area measured on ATom is dominated by the accumulation mode (b). Initial $SO_2$=1 pptv, and initial organics=7 pptv.

At line 470-477, you say that variation in the background aerosol doesn't help the Vehkamaki et al nucleation scheme, which is the scheme that is best constrained by your measurements. But in Figure S52 I see some blue dots. Perhaps (if I'm not wrong) you could say that if the CS were smaller by some factor, the Vehkamaki scheme could explain the observations, or could get reasonably close (NME=0.14), as shown in Figure S52? A NME of 0.14 for Vehkamaki might be just as 'good' as a lower NME for the schemes that include organics and NH3, since these vapors are not constrained by your measurements. Can you consider promoting this figure, and perhaps S51 for the RIC scheme, to the main text, as well as improving the caption (to refer to Table 1 where the various abbreviations are defined) and including a more extensive discussion?

The blue dots in Figure S52 correspond to the VEHK nucleation scheme with organics added for the initial particle growth. The best NME=0.14 here was obtained for $SO_2$=13.9 pptv and organics=7.2 pptv when the measured size distribution ($dN/dlog_{10}D_p$) >8 nm (instead of standard setting of $dN/dlog_{10}D_p$ >12 nm) was used as initial background aerosol to initiate the simulation.

We have added the following text to the manuscript, line 481:
"Varying the pre-existing initial aerosol or completely removing background particles in the VEHK scheme with added organics may improve the fits for certain initial conditions (Fig.S52), making it plausible for better NME values if $CS_{>12nm}$ was $4.12x10^{-5}$ $s^{-1}$, and SO2 and organics were 13.9 pptv and 7.2 pptv, respectively, for this particular case. Out of 32 cases studied here, there is no case with NME <0.2 for VEHK scheme. Once we add organics for the initial growth the NME improves resulting in 6 cases with NME<0.2 (Table S4)."

The caption has been updated and reads now: "Figure S52: The effect of varying the initial background aerosol on the NME in TOMAS for case sd486 (ATom2, 2017-02-03, 03:05:31-03:06:30 UTC) and VEHK nucleation scheme and VEHK with organics added for initial growth. OH at solar zenith angle of $0^0$ was set to $3x10^6$ molec $cm^{-3}$. SD refers to the number size distribution $dN/dlog_{10}D_p$. Table 1 in the main manuscript describes the abbreviations used here."

**Other minor comments**
A mention of the update to the Vehkamaki (2002) nucleation scheme by Maattanen et al (JGR, 2017; https://agupubs.onlinelibrary.wiley.com/doi/full/10.1002/2017JD027429) is warranted. The update is valid over a wider range of temperature and humidity, which is relevant here. The new model is also validated against CLOUD measurements, includes ion-induced nucleation, and updated quantum chemistry calculations. Similarly, the ternary H2SO4-NH3 scheme of Yu et al, GMD 2020, has look-up tables ready for a model implementation: https://gmd.copernicus.org/articles/13/2663/2020/gmd-13-2663-2020.pdf . I think there is no need to update the schemes used for this paper, but perhaps they could be mentioned as worth exploring in future work?

In Section 3.3. Discussion line 647 in the revised manuscript we added (in italic here) to the existing text: "We also have not explored the organic-only nucleation scheme described by Kirbky et al. (2016), *an updated version of the Vehkamaki et al. (2002) scheme covering a wider range of temperatures and relative humidities by Maattanen et al. (2017) that has been validated against*

*CLOUD measurements, or the recently published ternary nucleation look up tables for model implementation (Yu et a., 2020), and these schemes are worth investigating in future studies."*

What motivates the night-time OH value of $1 \times 10_5$? Production from ozonolysis of terpenes? It seems like quite a high value to me.

We have chosen a low non-zero value for night-time OH in TOMAS based on our best estimate at the time. There is no night-time data on OH for the tropical upper troposphere during ATom, and $1 \times 10^5$ molec $cm^{-3}$ is the approximate detection limit of the instrument. The night-time OH value of $1 \times 10^5$ molec $cm^{-3}$ is an order of 10x smaller than the day-time values and the nucleation and growth of particles simulated in this study come almost entirely from the day-time. Thus, lowering the night-time value further should not change the result.

It is stated that trajectories generally stayed at high altitudes (line 387). How does the trajectory model handle deep convection? Presumably it is parameterized? Your Figure 3 shows some evidence that some trajectories do originate in the boundary layer, which seems reasonable, but are there not some big uncertainties here even with the ensemble approach adopted? It would be worth referencing and perhaps discussing Andreae et al (2018) (the paper in ACP, already cited) in this context, as they do a similar trajectory analysis in their paper.

The Traj3D model does not include an explicit convective parameterization. It uses the vertical velocities from the input reanalysis fields, in this case 0.25 deg resolution NCEP GFS, which parameterizes subgrid convection. Thus, the trajectories may not simulate the rapid transport that a specific parcel might undergo in a convective event, but rather to capture the general upward motion at the reanalysis scale.

We added the following text in line 402: "Similar trajectory analyses, in terms of examining the history of the sampled air masses for interactions with deep convection, have been undertaken by Froyd et al. (2009) and Andreae et al. (2018)."

Line 424 Atlantic is misspelt.
corrected

How does RH vary with time since convection? I'm guessing it doesn't vary very much and so is not super-important, but would be good to confirm this. I note it is represented in your simulations in any case (line 324)

Yes, that is correct. The RH along the trajectory is represented in our simulations. The shorter the time since convective influence (CI) the smaller the change in RH between $t_0$ (start of the trajectory, at the point air left the cloud) and $t_{fin}$ (at the point of aircraft location/measurement) (Fig.S67 below). We added Figure S67 to the SM.

[Figure]

Fig.S67. The difference in relative humidity (RH) along the trajectory between the trajectory location at the cloud edge ($t_0$) and the point of the aircraft location ($t_{fin}$) for 32 simulated cases.

We added the following text in the manuscript line referenced above: "The change in RH along the trajectory between the trajectory location at the cloud edge ($t_0$) and the point of the aircraft location ($t_{fin}$) for 32 simulated cases is presented in Fig. S67."

Supplement S2, line 457, and Figure 6: Could the rather strange size-bin emptying behaviour be avoided by shortening the timesteps of the box model?

The bin emptying will not go away with a shorter timestep, and it is a common feature of 2-moment microphysics schemes that use a top-hat or moving-center approach for condensational growth. We have added these above sentences to the SM (line 457).

Also, is size bin emptying actually responsible for the rather strange kinks in the size distribution frequently observed around 10nm, and which correspond to a much smaller kink in the observed size distributions? In Figure S25, this is especially apparent. The gaps in the size distribution are far larger than a single bin. Do you have some ideas for what might be happening here?

The gap is from zeroing out the nucleation mode in the initial size distribution. We start with no particles below 12 nm in the simulations, and the new particles have not grown past a few nm, creating a gap between a few nm and 12 nm.

The blue line in Figure S25 represents output from the MAIA model, which does not experience the issues with bin emptying. We do not apply any smoothing in MAIA. In MAIA we use surface area of the Aitken and accumulation mode as initial background aerosol (size distribution on the right in the plot). Nucleation mode particles that appear in the simulations have not grown yet to bigger sizes, and that is the reason for the gap between these modes.

L618: "Nucleation schemes are simply imperfect parameterizations extrapolated from laboratory observations": this is a bit too much generalization; it's not quite true for Vehkamaki (2002) for example. On the other hand, the RIC scheme you prefer to use is one of the more uncertain parameterizations from the CLOUD experiment, because the organic molecules thought to participate in nucleation were not measured directly in the chamber when that parameterization was developed. For example, in the conclusions, perhaps it's worth pointing out that it is not surprising that one of the least well constrained nucleation mechanisms – the RIC scheme, which is the one you can tune most easily by changing the organic concentration – can be made to agree with observations the best.

We deleted this sentence and updated the sentence in line 713 by additional text: "Also, the Riccobono scheme*, one of the least constrained nucleation mechanisms,* required a large extrapolation in temperature to simulate UT conditions."

L1037 several references are merged together.

Corrected.

/////////////////////////////////////////////////////////////////////////////////////////////////////////////////////////////////////

**Response to the Anonymous Referee #2**

The authors thank the reviewer for helpful comments that have improved the manuscript. The discussion below contains the original text by the reviewer and our responses (in blue) along with changes made to the revised manuscript. Additional minor changes to the manuscript are indicated at the end of this document.

This manuscript tackles a globally important research topic: formation of new aerosol particles in the tropical upper troposphere. The paper is essentially a sensitivity study, aiming to give new insight into which nucleation mechanisms and aerosol precursors, coupled with the initial growth of newly-formed particles, best explain the observed ultrafine particle number size distribution. The conducted analysis is based on box model simulations and statistical analyses of the simulation results. The paper is scientifically sound and relatively well structure. I do have, however, a few issues that should be addressed before the paper is ready for publication.

My major criticism is related to the treatment of aerosol processes in convective outflow regions. As the authors state, they simulate outflow regions of deep convective clouds using a box model.

This is fine as long as both nucleation and growth occur well beyond the region abound cloud boundaries where most of the mixing between cloud outflow and upper troposphere air take place. This may not be the case, as it is quite possible that nucleation and early particle growth take place in the mixing region, or event inside the convective clouds. This would seriously bias the results obtained in the paper. I understand that including the cloud and its immediate outflow region in a box model is almost impossible, and therefor outside the scope of the current paper. However, the authors should bring up this issue more honestly as done in the present paper (brief mentioning on lines 628-629). Furthermore, there are a number of both modeling and observations studied conducted on new particle formation in cloud outflow regions. The authors should better acknowledge such studies when discussion their results, their implications and the associated uncertainties.

More details have been added in section: 3.3 Discussion, line 661:
"The box model used here simulates NPF in the outflow region of deep convective clouds. Although, active NPF was identified in the vicinity of clouds and in the cloud outflow region in many studies (such as Perry and Hobbs 1994, Clarke et al., 1998, 1999; Ström et al., 1999; Clement et al. 2002; Twohy et al., 2002; Weigelt et al., 2009; Waddicor et al., 2012; de Reus et al., 2001; Clarke and Kapustin 2002; or Andreae et al., 2018), the exact location of NPF with respect to cloud remains uncertain (Kulmala et al., 2006; Waddicor et al., 2012). We assume that nucleation does not occur within the cloud, and that the outflow does not immediately mix with the surrounding air in the highly stratified upper troposphere. If NPF were to occur within the cloud or in a zone of turbulent mixing at the cloud edge, as suggested by some studies (Ström et al., 1999; Lee et al., 2004; Weigel, 2011; Kazil et al., 2007; Kulmala et al., 2006), our results would be biased."

The references mentioned in the above paragraph were added to the list of references.

Comment: Other, minor issues:
The right parenthesis is missing from line 432.
Corrected.

The text on lines 538-543 is not logical. When discusses nucleation mechanism not involving NH3 at all, it is incorrect to say "regardless of NH3 concentrations", as the outcome of such mechanisms does not depend on NH3 concentrations. Please correct.

Thank you for noticing that. We have corrected this sentence and moved it to lines 545-547. The sentence reads now: "Further, regardless of the available $SO_2$, the results strongly suggest that binary sulfuric acid-water nucleation, whether ion-assisted or neutral, and whether coupled with organic growth or not, generally cannot explain the ATom observations."

The statement on lines 546-548 sounds a bit strange. Is really so that the concentration of both SO2 and organic precursors need to be smaller than some upper limit values to reproduce the observations? Please check out this statement and modify if needed.

Our results show that in order to reproduce observations, the initial conditions of $SO_2$ were found to be within the range of mixing ratios measured on ATom 4 and in other studies (Rollins et al., 2017, 2018). While the initial conditions of organics, not measured on ATom, were found to be

within the chosen plausible range of initial mixing ratios. We needed to set the lower and upper limit for the initial conditions used in simulations and tried to make sure their values were either based on measurements or literature, and if the species were not measured, that the limits are at least plausible.

The $SO_2$ and $NH_3$ mixing ratios were varied between 1 and 100 pptv to explore a large range of plausible conditions. The evaluated $SO_2$ range exceeds that measured on ATom 4 (Supplemental Material Figs. S4 and S5) and covers the <30 pptv mixing ratios reported in other studies in the UT over the central and western tropical Pacific (Thornton et al., 1997; Rollins et al., 2017, 2018). Organic aerosol precursors are unknown in the UT and were not directly measured; thus, we explored a range of probable mixing ratios between 1 and 100 pptv (of precursors with an SOA yield of 1).

We highlight that the simulated size distributions that show good agreement with observations (NME< 0.2) are characterized by rather low initial mixing ratios of $SO_2$. These low mixing ratios are within the range of $SO_2$ mixing ratios measured during ATom (< 30 pptv) at these locations.

We state that: "We note that we performed no simulations with mixing ratios of $SO_2$ or organics above 100 pptv. While we cannot exclude that for some cases the mixing ratios of these precursors at levels above 100 pptv could improve fits, these levels are outside of prior observations so were not considered in this study."

We modified the statement as follows: "The findings for the case of organic-mediated NPF are summarized in Fig. 8, where we show the $SO_2$ and organic precursor mixing ratios for all sensitivity simulations using the RIC scheme, highlighting the assumptions that yielded the lowest NME for each case."

////////////////////////////////////////////////////////////////////////////////////////////////////////////////////////////////////////////////////////////////

**Response to the Anonymous Referee #3**

The authors thank the reviewer for helpful comments that have improved the manuscript. The discussion below contains the original text by the reviewer and our responses (in blue) along with changes made to the revised manuscript. Additional minor changes to the manuscript are indicated at the end of this document.

The paper investigates the origin of nucleation mode particles in the upper tropical troposphere and makes use of the extensive ATom dataset in the process. The authors use nucleation and growth box models evaluated along the trajectories of air masses reaching from the outflow of tropical convection until the air was encountered during ATom flights. They find that the nucleation parametrization of Riccobono et al. 2014can describe the measured datasets best, which indicates organic involvement in nucleation. Also, if growth driven by organics is included in the model, the agreement with observations is enhanced as well, further pointing to the importance of organics in the upper tropical troposphere.

The paper offers very interesting insights into the important, but notoriously hard to observe topic of upper tropospheric new particle formation. The authors corroborate their conclusions with an extensive set of measurement and model data that are nicely presented. The paper is well within the scope of ACP; it is well written and clearly structured. The methods used are described clearly as well as their limitations. I recommend publication in ACP and only have some minor comments to add:

The paper focuses on nucleation rates, which is very insightful, however, in the model also growth, especially driven by organics is incorporated. You state that the inclusion of organic growth enhances the agreement between model and observations in many cases. However, you do not give a range for the growth rates needed to do that. Can you add a figure and/or short discussion that indicate the range of growth rates used in your model?

We calculated growth rates (GR) for all cases and all schemes based on the time of first appearance and the diameter of the leading edge of the nucleation mode. We used a threshold concentration for where the leading edge is when $dN/dlog_{10}D_p > 10$ $cm^{-3}$ in any of the size bins smaller than 12 nm. The growth rates were not calculated for cases with weak nucleation events i.e. with $dN/dlog_{10}D_p$ never exceeding 10 $cm^{-3}$ in any of the size bins smaller than 12 nm.

The growth rates for the RIC scheme are presented in Figure S71 below.

The following text was added to line 513:" Growth rates calculated basing on the diameter of the leading edge and threshold value of $dN/dlog_{10}D_p > 10$ $cm^{-3}$ in any of the size bins below 12 nm were mostly between 0.1 and 3 nm $hr^{-1}$ for the RIC scheme (Figure S71). The growth rates for all cases investigated here using RIC and other schemes are presented in Figure S72."

[Figure]

Figure S71. Growth rates calculated for simulations with $SO_2$ that have the lowest (best) NME for each case (32 cases in total) studied using RIC scheme. Calculation is based on the diameter of the leading edge and threshold value of $dN/dlog_{10}D_p > 10$ cm$^{-3}$ in any of the size bins below 12 nm.

[Figure]

Figure S72. Growth rates calculated for the leading-edge diameter and threshold value of $dN/dlog_{10}D_P >10$ $cm^{-3}$ in any of the size bins below 12 nm for simulations with $SO_2$ that have the lowest (best) NME in (a) VEHK and with added organics for the initial growth of the particles, (b) NAPA and with added organics, (c) NAPAt and with added organics, (d) RIC, (e) DUN NH3=0 and with organics added for initial growth, and (f) DUN and with organics added. Note different y-axis ranges in (a,b,c) and (d,e,f). Blue and red symbols represent simulations without and with organics added for initial growth of the particles, respectively. The RIC scheme is the only one here that includes organics both in the particle nucleation and growth.

Are you in Table 1 discussing the ion concentration as indicated by the given parameter name or the ion pair production rate as indicated by the given unit (cm-3s-1)? Please add a short description in section 2.4 that describes the origin of this quantity. The value 15 in Table 1 would correspond to the ion pair production rate in cm-3s-1 given in Dunne et al. 2016 for the upper tropical troposphere.

Thank you for noticing. It has an ion pair production rate of 15 $cm^{-3}$ $s^{-1}$, which is typical for the upper tropical troposphere as given in Dunne et al. (2016). We changed the text from 'ion concentration' to 'ion pair production rate' in Table 1 and added a footnote under it: "value typical for the tropical upper troposphere (Dunne et al., 2016)"

l. 322: You state "We have undertaken sensitivity studies that vary the pre-existing background aerosol used as input parameter (Table 1)", but in Table 1 there is no information on how this quantity was varied, as it is only named. Can you give more detail in Table 1 on how you varied the pre-existing aerosol and/or refer to the SI part where you discuss this in more depth?

The pre-existing background aerosol in the table refers to the "size distribution" as listed in the table (row below). We modified the table entry to make it more straightforward and combined the name of this parameter: "Background pre-existing aerosol: initial input size distribution (SD)". In the table's footnote it is mentioned how this parameter was varied: "initial background aerosol size distribution was varied: SD>12 nm means background SD as described in the text was used to initiate the model; SD>12nm x2 means background SD multiplied by 2; SD>12nm /2 means SD divided by 2; SD=0 means no background aerosol; SD>12 nm-5 nm means SD was shifted by 5 nm to smaller diameters; SD>8nm means measured background SD >8nm was used as initial SD."

We added: "where SD refers to the number size distribution $dN/dlog_{10}D_p$".

l. 485: You state that "varying the scale factor for organics taking part in nucleation (Forgnuc) did not change results significantly". Can you add a short discussion on why this is the case? You span one order of magnitude in [BioOxOrg] with the scaling factor values you use, so according to the RIC scheme, this should vary the nucleation rate as well by an order of magnitude. So in Fig S48 I would guess that you would need higher organics in the F = 0.1 case to match observations than in the F = 1 case. However, the blue dots are practically identical in all cases. So what is the compensating effect for that?

Nucleation can have strong self-regulating mechanisms, and these are likely contributing here. Increasing $F_{orgnuc}$ increases the nucleation rate initially in the high- $F_{orgnuc}$ simulations, which increases the condensation and coagulation sinks, which decreases the vapor concentrations (and hence nucleation and growth rates), and increases the coagulational loss rates. Each of these feedbacks lead to a buffering of changes to the nucleation mode. Westervelt et al., (2014) describes these buffering mechanisms in detail. Changing $F_{orgnuc}$ does not change the amount of organic material and sulfuric acid that condenses though, so ultimately the size distributions are relatively insensitive to changing $F_{orgnuc}$ because the buffering mechanism and total condensation remain approximately constant.

We added the text above verbatim to line 488.

Figure 7: It is a bit misleading that you place the black symbols for Aitken and accumulation mode at x = 0, as these originated from the measurement as you write and the measurement was not taken at x = 0. Even if you use them as starting point for the model, I would still place them at the point of actual measurement.

We have moved these to the point of actual measurement.

Figure 7: I would suggest to remove grid lines in between the panels.

Done.

l. 577: "5-20 size range": You mean nm?

Yes. Corrected now.

Figure 2: Add an altitude axis on the right, as in Fig 2 in Williamson et al. 2019

Done. The updated Figure 2 is presented below. We updated Figure S6 for Atlantic in the SM accordingly.

[Figure]

Figure 2: Ambient pressure as a function of latitude colored by the measured number concentration of particles with $D_p$ from 3 - 60 nm over the Pacific Ocean for a) ATom 1, July-August 2016; b) ATom 2, January-February 2017; c) ATom 3, September-October 2017; and d) ATom 4, April-May 2018). Periods of flight in clouds, over continents and near airports have been removed.

The following references have been added to the manuscript:

Clarke, A. D., F. Eisele, V. N. Kapustin, K. Moore, D. Tanner, L. Mauldin, M. Litchy, B. Lienert, M. A. Carroll, and G. Albercook, Nucleation in the equatorial free troposphere: Favorable environments during PEM-Tropics, J. Geophys. Res., 104, 5735–5744, 1999.

Clement, C. F., I. J. Ford, C. H. Twohy, A. J. Weinheimer, and T. Campos, Particle production in the outflow of a mid-latitude storm, J. Geophys. Res., 10.1029/2001JD001352, 2002.

de Reus, M., R. Krejci, R. Scheele, J. Williams, H. Fischer, and J. Ström, Vertical distributions of the aerosol number concentration and size distribution over the northern hemisphere Indian Ocean, J. Geophys. Res., 106, 28,629–28,642, 2001.

Lee, S.-H., et al., New particle formation observed in the tropical/subtropical cirrus clouds, J. Geophys. Res., 109, D20209, doi:10.1029/2004JD005033, 2004.

Perry, K. D., and P. V. Hobbs, Further evidence for particle nucleation in clear air adjacent to marine cumulus clouds, J. Geophys. Res., 99,22,803–22,818, doi:10.1029/94JD01926, 1994.

Ström, J., H. Fischer, J. Lelieveld, and F. Schröder, In situ measurements of microphysical properties and trace gases in two cumulonimbus anvils over western Europe, J. Geophys. Res., 104, 12,221 – 12,226, 1999.

Waddicor, D. A., Vaughan, G., Choularton, T. W., Bower, K. N., Coe, H., Gallagher, M., Williams, P. I., Flynn, M., Volz-Thomas, A., Pätz, H.-W., Isaac, P., Hacker, J., Arnold, F., Schlager, H., and Whiteway, J. A.: Aerosol observations and growth rates downwind of the anvil of a deep tropical thunderstorm, Atmos. Chem. Phys., 12, 6157–6172, https://doi.org/10.5194/acp-12-6157-2012, 2012.

Weigelt, A., Hermann, M., van Velthoven, P. F. J Brenninkmeijer, C. A. M., Schlaf, G. and co-authors. Influence of clouds on aerosol particle number concentrations in the upper troposphere. J. Geophys. Res. 114, D01204, doi:10.01210.01029/02008JD009805, 2009.

Yu, F., Nadykto, A. B., Luo, G., and Herb, J.: H2SO4–H2O binary and H2SO4–H2O–NH3 ternary homogeneous and ion-mediated nucleation: lookup tables version 1.0 for 3-D modeling application, Geosci. Model Dev., 13, 2663–2670, https://doi.org/10.5194/gmd-13-2663-2020, 2020.

/////////////////////////////////////////////////////////////////////////////////////////////////////////////////////////////////////////

Additional minor changes to the manuscript / supplemental material:

Case sd82: typo in Table S4: NME of 0.07 corrected to 0.27 for the RIC scheme. This case has been recalculated for all schemes. All relevant tables and plots for this case were updated. This resulted in a change of the number of cases with best NME for the RIC from 22 to 21. The NAPA+org$_{gr}$ scheme has now 4 best NME cases rather than 3.

Figure 6a and figures S2, S14-46, S50, S53, subfigure (a) in the SM: y-axis label changed from dNDlogDp to $dN/dlog_{10}D_p$.

Figure S38: typo in a table: DUN+orggr changed from 1 to 100 pptv.